# Trained Transformer Classifiers Generalize and Exhibit Benign Overfitting In-Context

**Spencer Frei**
UC Davis*
sfrei@google.com

**Gal Vardi**
Weizmann Institute of Science
gal.vardi@weizmann.ac.il

## Abstract

Transformers have the capacity to act as supervised learning algorithms: by properly encoding a set of labeled training ("in-context") examples and an unlabeled test example into an input sequence of vectors of the same dimension, the forward pass of the transformer can produce predictions for that unlabeled test example. A line of recent work has shown that when linear transformers are pre-trained on random instances for linear regression tasks, these trained transformers make predictions using an algorithm similar to that of ordinary least squares. In this work, we investigate the behavior of linear transformers trained on random linear classification tasks. Via an analysis of the implicit regularization of gradient descent, we characterize how many pre-training tasks and in-context examples are needed for the trained transformer to generalize well at test-time. We further show that in some settings, these trained transformers can exhibit "benign overfitting in-context": when in-context examples are corrupted by label flipping noise, the transformer memorizes all of its in-context examples (including those with noisy labels) yet still generalizes near-optimally for clean test examples.

## 1 Introduction

A key feature of transformer-based large language models (LLMs) is their ability to perform *in-context learning*: by providing a few labeled examples to a trained transformer with an unlabeled example for which the user wants a prediction, LLMs can formulate accurate predictions without any updates to their parameters (Brown et al., 2020). This ability to perform in-context learning is influenced by the interplay between tokenization of inputs (i.e. how to feed data into the transformer), pretraining datasets, optimization algorithms used for pre-training, the particular architecture of the transformer, as well as the properties of the in-context examples that are provided at test time.

A number of recent works have sought to develop a deeper theoretical understanding of in-context learning by investigating transformers trained on classical supervised learning tasks. This line of work was initiated by the experiments of Garg et al. (2022), who showed that when transformers are trained on random instances of supervised learning problems, they learn to implement supervised learning algorithms at test-time: e.g., when training GPT2 architectures (Radford et al., 2019) on random linear regression tasks with weights sampled from a Gaussian prior, the transformer learns to implement an algorithm similar to ordinary least squares. This led to a series of follow-up works which analyzed the dynamics of gradient descent/flow over simplified linear transformer architectures when trained on linear regression tasks (von Oswald et al., 2022; Akyurek et al., 2022), some providing guarantees for how many tasks or samples were needed to generalize well (Zhang et al., 2024; Ahn et al., 2023; Wu et al., 2024).

In this work, we analyze the behavior of linear transformer architectures which are trained by gradient descent on the logistic or exponential loss over random linear classification tasks. We consider a restricted linear attention model, a setting considered in prior works (Wu et al., 2024; Kim et al., 2024). We assume that pre-training tasks are sampled from random instances of class-conditional Gaussian mixture model data, i.e. for some $R > 0$, covariance matrix $\Lambda$ and for each

---

*Now at Google DeepMind

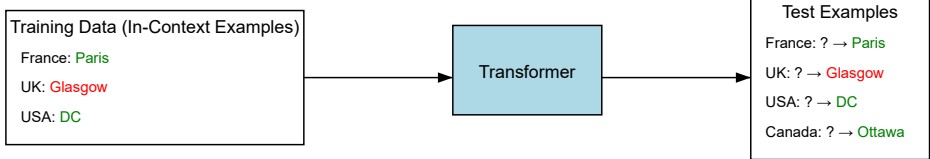

Figure 1: Benign overfitting in-context: after pre-training (not shown), when given a sequence of in-context examples, the transformer memorizes noisy labels yet still generalizes well to unseen test examples.

task $\tau = 1, \ldots, B$ we have

$$\mu_\tau \overset{\text{i.i.d.}}{\sim} \mathsf{Unif}(R \cdot \mathbb{S}^{d-1}), \quad y_{\tau,i} \overset{\text{i.i.d.}}{\sim} \mathsf{Unif}(\{\pm 1\}), \quad z_{\tau,i} \overset{\text{i.i.d.}}{\sim} \mathsf{N}(0, \Lambda), \qquad x_{\tau,i} = y_{\tau,i}\mu_\tau + z_{\tau,i}.$$

We assume that test-time in-context examples also come from class-conditional Gaussian mixture models as above but with two important differences. First, the signal-to-noise (determined by $R$) at test time can differ from those seen during pre-training. Second, we allow for label-flipping noise to be present at test time, i.e. we flip labels $y_i \mapsto -y_i$ with probability $p$.

Our first contribution is an analysis of how many pre-training tasks are needed in order to generalize well at test-time. We find that after pre-training the transformer can tolerate smaller SNR than those tasks which it was trained on. Moreover, we find that at test time the transformer can tolerate label-flipping noise, even though the pre-training data does not have label-flipping noise. Thus, even when pre-training on simple and easy-to-learn datasets, the transformer can generalize on more complex tasks. Our proof follows by an explicit analysis of the implicit regularization due to gradient descent during pre-training. To our knowledge, this is the first theoretical result on in-context learning for linear classification.

Our second contribution is the finding that the trained transformer can exhibit "benign overfitting in-context": namely, if the test-time sequence is subject to label-flipping noise, then in some settings the transformer memorizes the in-context training data yet still generalizes near-optimally. The precise setting where this occurs requires the features to lie in a high-dimensional space and for the SNR to be relatively small, a set of conditions observed in prior works on benign overfitting in classification tasks (Chatterji & Long, 2021; Frei et al., 2022; 2023a). Figure 1 illustrates what this phenomenon would look like in the language setting. To the best of our knowledge, no prior work demonstrated that transformers could exhibit benign overfitting.

## 2 RELATED WORK

**In-context learning for supervised learning tasks.** Following the initial experiments of Garg et al. (2022), a number of works sought to understand what types of algorithms are implemented by transformers when trained on supervised learning tasks. Akyurek et al. (2022) and von Oswald et al. (2022) used approximation-theoretic and mechanistic approaches to understand how a variety of transformer architectures could implement linear regression algorithms. Bai et al. (2024) showed that transformers could implement a variety of classical algorithms used for different supervised learning tasks, e.g. ridge regression, Lasso, and gradient descent on two-layer networks. Zhang et al. (2024), Ahn et al. (2023) and Mahankali et al. (2024) examined the landscape of single-layer linear transformers trained on linear regression tasks, with Zhang et al. (2024) additionally developing guarantees for convergence of (non-convex) gradient flow dynamics. There are other works which examine training of transformers on data coming from hidden Markov chains (Edelman et al., 2024) or nonparametric function classes (Kim et al., 2024). The most closely related work to this one is Wu et al. (2024), who focused on a convex linear transformer architecture, as we do in this work, trained on linear regression tasks with SGD. They provided task and (in-context) sample complexity guarantees when training by SGD. In contrast, we develop guarantees for the classification setting, and we develop guarantees via an analysis of the implicit regularization of GD.

**Implicit regularization in transformers.** The starting point for our analysis of the task complexity of pre-training and the sample complexity of in-context learning is an analysis of the implicit regularization of the optimization algorithm. We outline the most related works here and refer the reader to the survey by Vardi (2022). The convex linear transformer architecture we study is linear in (the vectorization of) its parameters, which by Soudry et al. (2018) implies that gradient descent converges in direction to the max-margin solution. The more general class of linear transformer architectures are non-convex in general, but certain subclasses of them are homogeneous in their parameters and hence converge (in direction) to KKT points of max-margin solutions (Lyu & Li, 2020; **?**). Another line of work seeks to understand the implicit bias of GD for softmax-based transformers (Ataee Tarzanagh et al., 2023; Tarzanagh et al., 2023; Thrampoulidis, 2024; Vasudeva et al., 2024), typically with more stringent assumptions on the structure of the training data.

**Benign overfitting.** The ability for neural networks to overfit to noise yet still generalize well (Zhang et al., 2017) led to a significant line of work on reconciling this phenomenon with classical learning theory. We focus on the most directly relevant works here and point the reader to the surveys Bartlett et al. (2021); Belkin (2021) for a more extensive review. Prior works have shown that kernel-based methods (Belkin et al., 2019) and the ordinary least squares solution (Bartlett et al., 2020) can exhibit benign overfitting in regression tasks, and that the max-margin solution can in classification tasks (Chatterji & Long, 2021; Frei et al., 2023a). Our proof technique comes from the works Frei et al. (2023c;a), where it was shown that the max-margin solution over 2-layer neural nets behaves similarly to a nearly uniform average of the training data and that this average can exhibit benign overfitting in class-conditional Gaussian mixture models. We similarly show that the forward pass of a pre-trained transformer behaves similarly to an average over the in-context training examples, and use this to show benign overfitting.

## 3 PRELIMINARIES

In this section we introduce the pre-training distribution, the particular transformer architecture we consider, and the assumptions on the pre-training data.

### 3.1 NOTATION

We denote matrices with capital letters $W$ and vectors and scalars in lowercase. The Frobenius norm of a matrix is denoted $\|W\|_F$, and its spectral norm as $\|W\|_2$. We use $a \wedge b := \min(a, b)$ and $a \vee b = \max(a, b)$. We use the standard $O(\cdot), \Omega(\cdot)$, and $\Theta(\cdot)$ notations, where $\tilde{O}, \tilde{\Omega}, \tilde{\Theta}$ ignore logarithmic factors. We use $\mathbb{1}(x)$ as the indicator function, which is 1 if $x$ is true and 0 otherwise. We denote $[M] = \{1, 2, \ldots, M\}$ for a positive integer $M$, and we denote $\mathsf{Unif}(R \cdot \mathbb{S}^{d-1})$ as the uniform distribution on the sphere of radius $R$ in $d$ dimensions. In Table 1 in the appendix, we collect all notation used throughout the paper.

### 3.2 SETTING

In the linear regression setting, there is a natural distribution of pre-training tasks that serves as a basis for a number of theoretical works on in-context learning in that setting (von Oswald et al., 2022; Akyurek et al., 2022; Zhang et al., 2024; Ahn et al., 2023), namely, a fixed distribution over the features $x$ and a Gaussian prior over the weights $w_\tau$ for each task $\tau$ and labels generated as $y = w_\tau^\top x$ per task. We are unaware of any prior works in the linear classification setting, so we propose the following. We assume each task is a random instance of a class-conditional Gaussian mixture model with cluster means $\mu_\tau \sim \mathsf{Unif}(R \cdot \mathbb{S}^{d-1})$. In particular, we assume the following.

**Assumption 3.1.** *Let $\Lambda \succ 0$ be a symmetric positive-definite $d \times d$ matrix and let $B, N \geq 1$ and $R > 0$. Let $\tau = 1, \ldots B$ and $i = 1, \ldots, N + 1$.*

1. *Let $\mu_\tau \stackrel{\text{i.i.d.}}{\sim} \mathsf{Unif}(R \cdot \mathbb{S}^{d-1})$ be uniform on the sphere of radius $R$ in $d$ dimensions.*

2. *Let $z_{\tau,i} \stackrel{\text{i.i.d.}}{\sim} \mathsf{N}(0, \Lambda)$, and $y_{\tau,i} \stackrel{\text{i.i.d.}}{\sim} \mathsf{Unif}(\{\pm 1\})$, where $\mu_\tau$, $z_{\tau,i}$ and $y_{\tau,i}$ are mutually independent.*

3. *Set $x_{\tau,i} := y_{\tau,i}\mu_\tau + z_{\tau,i}$.*

We tokenize samples to be fed into the transformer in the following way: for a sequence of $N$ labeled examples $(x_i, y_i)_{i=1}^N$ which we use to formulate a prediction for an unlabeled test example $x_{N+1}$, we concatenate the $(x_i, y_i)$ pairs into a $d+1$ dimensional vector and tokenize the test example as $(x_{N+1}, 0)$,

$$E = \begin{pmatrix} x_1 & x_2 & \cdots & x_N & x_{N+1} \\ y_1 & y_2 & \cdots & y_N & 0 \end{pmatrix} \in \mathbb{R}^{(d+1) \times (N+1)}. \tag{1}$$

In the standard softmax-based attention with a single head (Vaswani et al., 2017), the transformer is defined in terms of parameters $W^V \in \mathbb{R}^{d_e \times d_e}$, $W^K, W^Q \in \mathbb{R}^{d_k \times d_e}$. If $\theta = (W^K, W^Q, W^V)$ then softmax attention computes

$$f(E; \theta) = E + W^V E \cdot \text{softmax}\left( \frac{(W^K E)^\top W^Q E}{\rho_{N, d_e}} \right),$$

where $\rho_{N, d_e}$ is a normalization factor which is not optimized (but which may depend on $N$ and $d_e$). In *linear* transformers, the softmax is replaced with the identity function. Following prior work (von Oswald et al., 2022; Zhang et al., 2024; Ahn et al., 2023), we merge the $K$ and $Q$ matrices into $W^{KQ} := (W^K)^\top W^Q$ and we consider an objective function where the aim is to use the first $N$ columns of (1) to formulate predictions for $x_{N+1}$, whereby the bottom-right corner of the output matrix of $f(E; \theta)$ serves as this prediction. Writing $W^\Delta = \begin{pmatrix} W_{11}^\Delta & w_{12}^\Delta \\ (w_{21}^\Delta)^\top & w_{22,}^\Delta \end{pmatrix}$ for $\Delta \in \{V, KQ\}$, for the linear transformer architecture, this results in the prediction

$$\widehat{y}(E; \theta) = \left( (w_{21}^V)^\top \quad w_{22}^V \right) \cdot \frac{1}{N} E E^\top \cdot \begin{pmatrix} W_{11}^{KQ} \\ (w_{21}^{KQ})^\top \end{pmatrix} x_{N+1}.$$

Due to the product of matrices appearing above, the resulting objective function is non-convex, which makes the analysis of its training dynamics complex. In this work, we follow Wu et al. (2024); Kim et al. (2024) and instead consider a convex parameterization of the linear transformer which results from taking $w_{21}^{KQ} = w_{21}^V = 0$ and setting $w_{22}^V = 1$. This results in the following prediction for the label of $x_{N+1}$,

$$\widehat{y}(E; W) = \left( \frac{1}{N} \sum_{i=1}^N y_i x_i \right)^\top W x_{N+1}. \tag{2}$$

That is, $\widehat{y}(E; W)$ corresponds to the predictions coming from a linear predictor trained by one step of gradient descent on the logistic or squared loss (initialized at 0) but where the parameter $W$ is a (learned) matrix preconditioner. Prior work by Zhang et al. (2024) showed that when a single-layer linear transformer architecture is trained by gradient flow on linear regression tasks, it learns a function of the same type, where the preconditioner was found to correspond to the inverse covariance matrix of the pretraining data.

For each task $\tau$, denote the embedding matrix $E_\tau$ as the one formed using labeled examples $(x_{\tau, i}, y_{\tau_i})_{i=1}^N$ from Assumption 3.1 as per (1). We consider linear transformers which are trained on the prediction of the last token using the logistic or exponential loss, namely

$$\widehat{L}(W) := \frac{1}{B} \sum_{\tau=1}^B \ell\big(y_{\tau, N+1} \cdot \widehat{y}(E_\tau; W))\big), \qquad \ell \in \{q \mapsto \log(1 + \exp(-q)), q \mapsto \exp(-q)\}.$$

We are interested in the behavior of gradient descent on this objective,

$$W_{t+1} = W_t - \alpha \nabla \widehat{L}(W_t).$$

Note that $\widehat{y}(E; W)$ is homogeneous in $W$ (linear in $\text{vec}(W)$) and so by Soudry et al. (2018) we know that gradient descent has an implicit bias towards maximum-margin solutions, as we recall in the following theorem:

**Theorem 3.2** (Soudry et al. (2018))**.** *Define the $\ell_2$-max margin solution as*

$$W_{\text{MM}} := \text{argmin}\left\{ \|U\|_F^2 : \left( 1/N \sum_{i=1}^N y_{\tau, i} x_{\tau, i} \right)^\top U y_{\tau, N+1} x_{\tau, N+1} \geq 1, \forall \tau = 1, \ldots, B \right\}. \tag{3}$$

*If there exists $U$ which satisfies the constraints in (3), then provided $\alpha$ is sufficiently small, then $W(t)$ converges in direction to $W_{\text{MM}}$, i.e. for some constant $c > 0$ we have $W_t/\|W_t\| \to c W_{\text{MM}}$.*

This theorem shows that the max-margin solution (3) characterizes the limiting behavior of gradient descent. In the remainder, we will focus on the max-margin solution and derive all of the generalization guarantees as a consequence of the properties of the max-margin solution.

We next introduce the assumptions we need to ensure that the max-margin solution is well-behaved enough that we can later derive in-context learning guarantees. We fix a probability threshold $\delta \in (0, 1)$, and we shall show that all of our results hold with probability at least $1 - \delta$ provided the absolute constant $C > 1$ appearing below is a large enough universal constant (independent of $d$, $B$, $\Lambda$, and $\delta$).

(A1) $R$ is large enough so that

$$R^2 \geq C^2 \sqrt{d \operatorname{tr}(\Lambda^2)} \vee C^2 \left( \frac{\operatorname{tr}(\Lambda)}{d} \vee \sqrt{\frac{\operatorname{tr}(\Lambda^2)}{N}} \vee \|\Lambda\|_2 \right) \log(2B/\delta)$$

(A2) For some $c_B > 0$, we have $B \geq c_B d$, and

(A3) $d \geq C \log^4(2B^2/\delta)$.

The first assumption (A1) guarantees that the signal-to-noise ratio is sufficiently large (recall that $\Lambda$ is the covariance of $z_{\tau,i}$ in the identity $x_{\tau,i} = y_{\tau,i}\mu_\tau + z_{\tau,i}$); provided $B$ is polynomial in $d$, this assumptions is satisfied for $\Lambda = I$ when $R \gg \sqrt{d}$. The second assumption (A2) specifies how many pre-training sequences we (pre-)train on; we shall see later on that the generalization error achieved for in-context examples is determined in part by how small $c_B \wedge 1$ is ($c_B \wedge 1$ larger ensures better generalization). Note that we allow $c_B$ to be non-constant (so that $o_d(d)$ tasks as permitted). The assumption (A3) is needed for a technical reason, namely to ensure certain concentration bounds regarding sub-exponential random variables. Finally, note the near-independence of these assumptions on the number of examples $N$ per task during training: in many situations (e.g., $\Lambda = I$), we only require a single demonstration (i.e., $N = 1$) of the form $(x_{\tau,1}, y_{\tau,1})$.

Finally, we introduce the distribution on the test-time in-context examples. We allow for a more general distribution at test-time than the one used during pre-training, where the in-context examples can have label-flipping noise and a potentially smaller cluster mean, corresponding to a potentially smaller signal-to-noise ratio. We assume, for $i = 1, \ldots, M$,

1. $\tilde{y}_i \sim \operatorname{Unif}(\{\pm 1\})$.
2. $y_i = \tilde{y}_i$ w.p. $1 - p$ and $y_i = -\tilde{y}_i$ w.p. $p$ (label flipping noise).
3. $\mu \sim \tilde{R} \cdot \operatorname{Unif}(\mathbb{S}^{d-1})$.
4. $z_i \sim \mathsf{N}(0, \Lambda)$ and $x_i = \tilde{y}_i\mu + z_i$.

We assume the random variables $\tilde{y}_i, \mu, z_i$ are all mutually independent. We allow for the size of the cluster mean $\tilde{R}$ at test-time to potentially be different than the size of the cluster mean $R$ from the pretraining data, and indeed we shall see in Theorem 4.1 below that we can permit a smaller $\tilde{R}$ than what we require for $R$. Also note that we allow for a potentially different sequence length at test-time ($M$) than was observed during training ($N$); prior work in the linear regression setting showed that the transformer's behavior can depend quite differently on $M$ vs. $N$ (Zhang et al., 2024).

## 4 MAIN RESULTS

### 4.1 GENERALIZATION

Recall that the transformer with parameters $W$ makes predictions by embedding the set $\{(x_i, y_i)\}_{i=1}^M \cup \{(x_{M+1}, 0)\}$ into a matrix $E$ and then using $\operatorname{sign}(\hat{y}(E; W))$ as our prediction for $y_{M+1}$. Our goal is to understand the expected risk of the max-margin solution, i.e. the probability of misclassification of the test example $(x_{M+1}, y_{M+1})$: using (2),

$$R(W_{\mathsf{MM}}) = \mathbb{P}_{(x_i, y_i)_1^{M+1}, \mu}(\operatorname{sign}(\hat{y}(E; W_{\mathsf{MM}})) \neq y_{M+1})$$

$$= \mathbb{P}\left( \left[ \frac{1}{M} \sum_{i=1}^M y_i x_i \right]^\top W_{\mathsf{MM}} y_{M+1} x_{M+1} < 0 \right). \tag{4}$$

Our main result regarding generalization is the following theorem.

**Theorem 4.1.** *Let $\delta \in (0,1)$ be arbitrary. Suppose that $p = 1/2 - c_p$ where $c_p \in (0, 1/2]$ is an absolute constant. There are absolute constants $C > 1, c > 0$ depending only on $c_p$ such that if assumptions (A1), (A2), and (A3) hold, then with probability at least $1 - 10\delta$ over the draws of $\{\mu_\tau, (x_{\tau,i}, y_{\tau,i})_{i=1}^{N+1}\}_{\tau=1}^B$, when sampling a new task $\{\mu, (x_i, y_i)_{i=1}^{M+1}\}$, the max-margin solution (3) satisfies*

$$\mathbb{P}_{(x_i, y_i)_{i=1}^{M+1}, \, \mu}\big(\mathrm{sign}(\widehat{y}(E; W_{\mathsf{MM}})) \neq y_{M+1}\big)$$

$$\leq (p + 2\exp(-cM))\,\mathbb{1}(p > 0) + 2\exp\left(-c\rho\sqrt{d}\right) + 4\exp\left(-\frac{c\rho\tilde{R}}{\|\Lambda^{1/2}\|_2}\right) + 2\exp\left(-\frac{c\rho M^{1/2}\tilde{R}^2}{\|\Lambda\|_2\sqrt{d}}\right),$$

*where $\rho := \frac{c_B \wedge 1}{\log^2(2B^2/\delta)}$.*

Let us make a few observations on the above theorem. For simplicity let us assume that the covariance matrix $\Lambda = I_d$, so that $\|\Lambda^{1/2}\| = \|\Lambda\| = 1$, and let us assume we want the results to hold with a fixed probability threshold of $\delta = 0.001$. Then assumptions (A1) through (A3) are satisfied for any $N \geq 1$ examples per task so long as $R = \|\mu_\tau\| \gg C\sqrt{d}$. For the test error, there are a number of different regimes which require different considerations.

- Due to the dependence on $\rho$, the test error degrades when $c_B$ gets smaller, and we require $c_B = \tilde{\Omega}(d^{-1/2})$ to achieve a non-vacuous generalization bound, i.e. $B = \tilde{\Omega}(d^{1/2})$ pre-training tasks suffices if $\tilde{R}$ and $M$ are large enough.[1]

- While we require pre-training signal $R = \Omega(\sqrt{d})$, to achieve test error near the noise rate at test-time the signal $\tilde{R}$ can be as small as $\tilde{R} = \tilde{\Omega}(1)$ provided $M$ is large enough and $\rho = \tilde{\Omega}(1)$. This means the signal-to-noise ratio at test-time can be significantly smaller than what was observed during pre-training.

- When $p = 0$, there is no label-flipping noise. Here it is possible to have as few as one example per task ($M = 1$) provided $\tilde{R} = \tilde{\Omega}(d^{1/4})$, and $\rho = \tilde{\Omega}(1)$.

- When $p > 0$, there is label-flipping noise. The same analysis holds as in the $p = 0$ setting except that we always require $M = \tilde{\Omega}(1)$, which makes intuitive sense (with label noise, one must see more than one example to guarantee generalization w.h.p.). Notably, there was no label noise during pre-training, yet the transformer can generalize under label noise at test time.

## 4.2 BENIGN OVERFITTING

In this subsection we shall assume that the covariance matrix $\Lambda = I_d$. This is needed for technical reasons of the proof. We could more accommodate a covariance matrix of the form $\Lambda$ where $c_1 I \preceq \Lambda \preceq c_2 I$ but we just present the case $\Lambda = I$ for simplicity.

Our goal will be to show that the max-margin transformer can exhibit *benign overfitting in-context*. By this we mean that in its forward pass, the transformer memorizes all of the in-context examples, yet still generalizes well for new test examples. To make this more precise, for a sequence of training examples $\{(x_i, y_i)\}_{i=1}^M$, denote $E(x)$ as the embedding matrix corresponding to the sequence $(x_i, y_i)_{i=1}^M, (x, 0)$; recall that the examples $(x_i, y_i)_{i=1}^M$ are used to formulate predictions for $x$. We say that the transformer exhibits benign overfitting in-context if,

- All training examples are memorized: with high probability (over $\mu$ and $(x_i, y_i)_{i=1}^M$), for each example $(x_k, y_k)$, $k \leq M$, we have $\mathrm{sign}(\widehat{y}(E(x_k); W)) = y_k$.

- Test examples are classified near-optimally: $\mathbb{P}_{\mu, \, (x_i, y_i)_{i=1}^{M+1}}(\mathrm{sign}(\widehat{y}(E(x_{M+1}); W) = y_{M+1}) \leq p + \varepsilon$ for some vanishing $\varepsilon$.

---

[1] We also require $B = O(\mathrm{poly}(d))$, this comes from using a union bound over the pre-training tasks to ensure the max-margin solution is well-behaved.

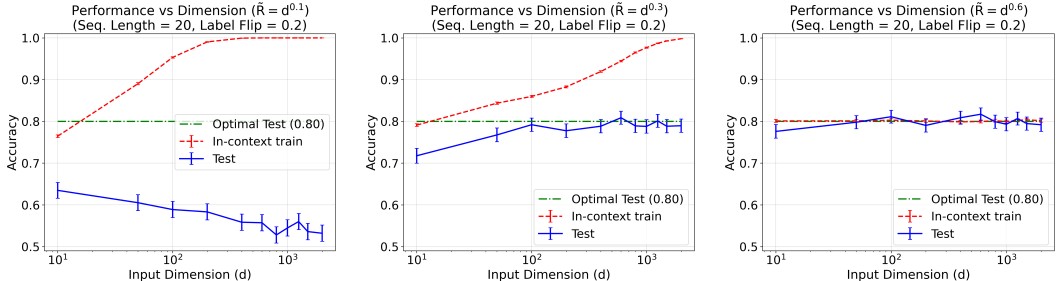

Figure 2: Cluster separation $\tilde{R}$ affects both (in-context) train and test accuracy, with small $\tilde{R}$ leading to overfitting, and large $\tilde{R}$ leading to better test accuracy; in between, benign overfitting occurs.

In Figure 1 we provide a schematic of what this phenomenon would look like in language tasks, where a user provides a sequence of country-capital pairs, some of which have errors, and the transformer repeats these errors at test time for those countries which it has seen with noisy labels, yet still generalizes well for never-before-seen countries.

To prove this phenomenon holds, we need to show both memorization and generalization. Theorem 4.1 in the previous subsection proved the generalization part, thus all that remains is to show that memorization can co-occur with generalization. The next theorem demonstrates that this is possible if the dimension $d$ is sufficiently large with respect to the number of in-context examples $M$.

**Theorem 4.2.** *Assume $\Lambda = I$ and let $\delta \in (0,1)$ be arbitrary. Suppose $p = 1/2 - c_p$ where $c_p \in (0, 1/2)$ is an absolute constant. There are absolute constants $C > 1$, $c > 0$ depending only on $c_p$ such that if assumptions (A1) through (A3) hold, then the following holds. With probability at least $1 - 10\delta$ over the draws of $\{\mu_\tau, (x_{\tau,i}, y_{\tau,i})_{i=1}^{N+1}\}_{\tau=1}^{B}$, for all $\tau \in [B]$, when sampling a new task $\{\mu, (x_i, y_i)_{i=1}^{M+1}\}$, the max-margin solution* (3) *satisfies,*

- *In-context training examples are memorized: for $\rho := \frac{c_B \wedge 1}{\log^2(2B^2/\delta)}$,*

$$\mathbb{P}_{(x_i, y_i)_{i=1}^{M}, \, \mu} \left( \exists k \in [M] \ \ s.t. \ \text{sign}(\widehat{y}(E(x_k); W_{\mathsf{MM}})) \neq y_k \right)$$
$$\leq 4M \exp\left( -\frac{c\rho\sqrt{d}}{\sqrt{M}} \right) + 8M \exp\left( -\frac{c\rho d}{M(\tilde{R}^2 \vee \tilde{R})} \right).$$

- *In-context test example achieves test error close to the noise rate:*

$$\mathbb{P}_{(x_i, y_i)_{i=1}^{M+1}, \, \mu} \left( \text{sign}(\widehat{y}(E(x_{M+1}); W_{\mathsf{MM}})) \neq y_{M+1} \right)$$
$$\leq p + 2\exp(-cM) + 2\exp\left( -c\rho\sqrt{d} \right) + 4\exp\left( -c\rho\tilde{R} \right) + 2\exp\left( -\frac{c\rho M^{1/2}\tilde{R}^2}{\sqrt{d}} \right),$$

The claim regarding generalization in the above theorem is the same as in Theorem 4.1 (we focus here on the case $p > 0$ since label noise is essential for *overfitting*), and the same comments following that theorem apply here as well. But a natural question is whether one can satisfy both memorization and near-noise-rate test error, i.e. benign overfitting. This is indeed possible, and can be seen most easily in the following setting: let $\delta = 0.001$, let $M$ be a large constant satisfying $p + 2\exp(-cM) \leq p + 0.0001$, and assume $B = d$ so $\rho = 1/\log^2(2d^2/\delta)$. If $\tilde{R} = d^\beta$ for $\beta \in (1/4, 1/2)$, then w.p. at least 99.9% over the pre-training data, since $\beta < 1/2$ memorization occurs with probability $1 - o_d(1)$, and since $\beta > 1/4$ the test error for the fresh test example $x_{M+1}$ is at most $p + 0.0001 + o_d(1)$. More generally, one can see that memorization occurs when $d \gg M(\tilde{R}^2 \vee \tilde{R})$ and $d \gg M$, i.e. when the dimension is large relative to the number of samples and when the signal of the test-time sequence is not too large relative to the dimension divided by the number of in-context samples.

While Theorem 4.1 and Theorem 4.2 apply for the infinite-time limit of gradient descent, a natural question is whether gradient descent trained for finite steps has similar behavior, and we indeed find

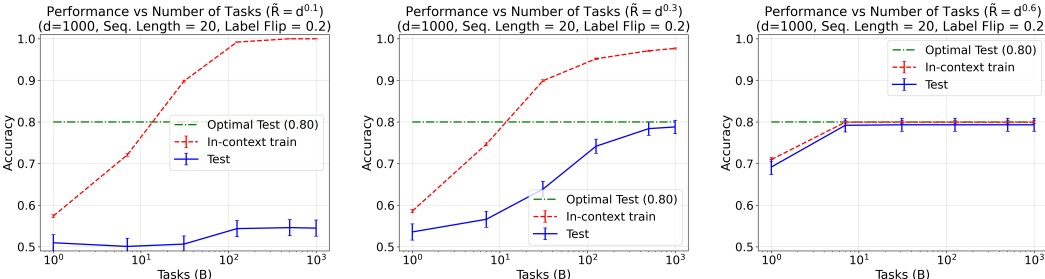

Figure 3: In-context test accuracy improves as the number of pre-training tasks (batch size) $B$ approaches the ambient dimension $d$, provided $\tilde{R}$ is large enough, with fewer $B$ needed for large $\tilde{R}$.

that it does. Figure 2 examines the role of high dimensionality and $\tilde{R}$, illustrating three qualitatively different phenomena: overfitting without generalization, when $\tilde{R}$ is small and $d/M$ is large; benign overfitting, when $\tilde{R}$ is in the sweet spot for generalization; and optimal test performance without overfitting, which occurs when $\tilde{R}$ is very large. In Figure 3, we show the effect of the number of pre-training tasks $B$ on the in-context test accuracy, again showing improved performance for more pre-training tasks with fewer pre-training tasks needed for optimal performance when $\tilde{R}$ is large, as suggested by Theorem 4.2. In both figures, we have GD-trained networks with step size $\eta = 0.01$ for 300 steps. Details on experiments and a link to our codebase are provided in Appendix E.

## 5    PROOF SKETCH

We will first discuss a sketch of the proof of generalization given in Theorem 4.1 when $\Lambda = I$. For notational simplicity let us denote $W = W_{\mathsf{MM}}$, $\widehat{\mu} := \frac{1}{M} \sum_{i=1}^{M} y_i x_i$, and let us drop the $M + 1$ subscript so that we denote $(x_{M+1}, y_{M+1}, \tilde{y}_{M+1}) = (x, y, \tilde{y})$. Then the test error is given by the probability of the event,

$$\{\mathrm{sign}(\widehat{y}(E; W)) \neq y_{M+1}\} = \{\widehat{\mu}^\top W y x \leq 0,\, y = \tilde{y}\} \cup \{\widehat{\mu}^\top W y x \leq 0,\, y = -\tilde{y}\}.$$

The test error can thus be bounded as

$$\mathbb{P}(\widehat{y}(E; W) \neq y_{M+1}) \leq \mathbb{P}(\widehat{\mu}^\top W \tilde{y} x \leq 0) + \mathbb{P}(y = -\tilde{y}) = \mathbb{P}(\widehat{\mu}^\top W \tilde{y} x \leq 0) + p.$$

Let us denote the examples $(x_i, y_i)$ in the test-time sequence which have clean labels $y_i = \tilde{y}_i$ as $i \in \mathcal{C}$, while those with noisy labels $y_i = -\tilde{y}_i$ as $i \in \mathcal{N}$. Then $\sum_{i=1}^{M} y_i x_i = (|\mathcal{C}| - |\mathcal{N}|)\mu + \sum_{i=1}^{M} y_i z_i = (M - 2|\mathcal{N}|)\mu + \sum_{i=1}^{M} y_i z_i$. If we denote $\widehat{p} := |\mathcal{N}|/M$ then by standard properties of the Gaussian, we have for $z, z' \overset{\text{i.i.d.}}{\sim} \mathsf{N}(0, I)$,

$$\widehat{\mu} \overset{\text{d}}{=} (1 - 2\widehat{p})\, \mu + M^{-1/2} z,$$

$$\tilde{y} x \overset{\text{d}}{=} \mu + z'.$$

We thus have

$$\mathbb{P}(\widehat{y}(E; W) \neq y_{M+1})$$
$$\leq p + \mathbb{P}\Big(\big((1 - 2\widehat{p})\mu + M^{-1/2} z\big)^\top W (\mu + z') < 0\Big)$$
$$= p + \mathbb{P}\Big((1 - 2\widehat{p})\mu^\top W \mu < -(1 - 2\widehat{p})\mu^\top W z' - M^{-1/2} z^\top W \mu - M^{-1/2} z^\top W z'\Big).$$

This decomposition allows for us to show the test error is near the noise rate $p$ if we can show that $(1 - 2\widehat{p})\mu^\top W \mu$ is large and positive while all of the other terms involving $\mu^\top W z'$, $z^\top W \mu$, and $z^\top W z'$ are small in comparison. Now if $\mu$ were a standard Gaussian, or if it were sub-Gaussian with independent components, then $\mu^\top W \mu$ would be close to its mean $\mathrm{tr}(W)$, with fluctuations determined from its mean determined by $\|W\|_F$, via a standard Hanson-Wright inequality. However,

since $\mu \sim \mathsf{Unif}(\tilde{R} \cdot \mathbb{S}^{d-1})$, the components of $\mu$ are not independent, and so we derive a modified version of Hanson-Wright (Lemma C.2) which applies in this setting where we show $\mu^\top W \mu$ is close to its mean $\frac{\tilde{R}^2}{d} \operatorname{tr}(W)$ with fluctuations from its mean determined by $\|W\|_F$. Thus, in order to show that the max-margin solution achieves (in-context) test error close to the noise rate $p$, we must show the following:

- A lower bound for $\operatorname{tr}(W)$, so that the mean of $\mu^\top W \mu$ is large and positive.

- An upper bound on $\|W\|_F$, so that we can control the fluctuations of $\mu^\top W \mu$ from its mean.

- An upper bound on $\widehat{p}$ so that $(1 - 2\widehat{p})$ is positive.

- Upper bounds on the quantities $|(1 - 2\widehat{p})\mu^\top W z'|$, $|M^{-1/2}z^\top W \mu|$ and $|M^{-1/2}z^\top W z'|$ so that these quantities are smaller than the positive term coming from the mean of $\mu^\top W \mu$. These, in turn, will require upper bounds on $\|W\|_F$.

Thus provided we have control over $\operatorname{tr}(W)$ and $\|W\|_F$, we can apply straightforward concentration inequalities to show that the (in-context) test error is near the noise rate. So let us now describe how to derive such bounds.

The starting place for our analysis is the definition of the max-margin solution (3). For notational simplicity let us define $\widehat{\mu}_\tau := \frac{1}{N} \sum_{i=1}^n y_{\tau,i} x_{\tau,i}$ and let us denote $x_{\tau,N+1}, y_{\tau,N+1}$ as $x_\tau, y_\tau$ so that the max-margin solution can be written

$$W := \operatorname{argmin}\{\|U\|_F^2 : \widehat{\mu}_\tau^\top U y_\tau x_\tau \geq 1, \ \forall \tau = 1, \ldots, B\}. \tag{5}$$

Since $\nabla \widehat{y}(E_\tau; W) = \widehat{\mu}_\tau x_\tau^\top$, the KKT conditions imply that there exist $\lambda_1, \ldots, \lambda_B \geq 0$ such that

$$W = \sum_{\tau=1}^B \lambda_\tau y_\tau \widehat{\mu}_\tau x_\tau^\top, \tag{6}$$

and moreover we have $\lambda_\tau = 0$ whenever $y_\tau \widehat{\mu}_\tau^\top W x_\tau \neq 1$. By a careful analysis of the KKT conditions (see Lemma B.1), we can derive nearly-matching upper and lower bounds for $\sum_\tau \lambda_\tau$, $\operatorname{tr}(W)$, and $\|W\|_F$: if we assume that $c_B > 0$ is an absolute constant (i.e. the number of tasks $B$ is greater than a constant multiple of the input dimension $d$) then these bounds take the form

$$\sum_{\tau=1}^B \lambda_\tau = \tilde{\Theta}(d/R^4), \qquad \operatorname{tr}(W) = \tilde{\Theta}(d/R^2), \qquad \|W\|_F = \tilde{\Theta}(\sqrt{d}/R^2). \tag{7}$$

These bounds, together with the concentration inequalities outlined above, suffice for proving Theorem 4.2.

As for the possibility of benign overfitting in-context shown in Theorem 4.2, since Theorem 4.1 guarantees generalization, the only task that remains to be shown is that *overfitting* occurs: namely, we want to ensure that for every example $(x_k, y_k)$, if we denote $E(x_k)$ as the embedding matrix with input sequence $(x_i, y_i)_{i=1}^M$ but with test example $x_k$, then we want to show that

$$\operatorname{sign}\left(\widehat{y}(E(x_k); W)\right) = y_k \iff \widehat{\mu}^\top W y_k x_k > 0.$$

Now, since $\operatorname{tr}(cI) = cd$ and $\|cI\|_F = c\sqrt{d}$, the identities in (7) suggest that $W$ has properties similar to that of a scaled identity matrix. If $W$ were indeed a scaled identity matrix, then the goal would be to show that

$$M\widehat{\mu}^\top y_k x_k = \left(\sum_{i=1}^M y_i x_i\right)^\top y_k x_k = \|x_k\|^2 + \sum_{i \neq k} \langle y_i x_i, y_k x_k \rangle \geq \|x_k\|^2 - \sum_{i \neq k} |\langle x_i, x_k \rangle| > 0.$$

That is, we would need to ensure that the examples $\{x_i\}_{i=1}^M$ are nearly-orthogonal in a particular sense (Frei et al., 2023c). This occurs provided the ambient dimension $d$ is much larger than the number of examples $M$ and the signal strength $\tilde{R}^2 = \|\mu\|^2$, and has been shown in prior work on benign overfitting in neural networks (Frei et al., 2022; 2023a). In our setting, $W$ is not a multiple of the identity matrix so this proof technique does not directly apply, but at a high level the proof ideas are similar, and appear in Section D.

# 6 CONCLUSION

In this work we developed task complexity and sample complexity guarantees for in-context learning class-conditional Gaussian mixture models with a single-layer linear transformer architecture. We analyzed the implicit regularization of gradient descent to characterize the algorithm implemented by the transformer after pre-training. This allowed for us to quantify how many in-context samples are needed in order to achieve small test error, which to the best of our knowledge has not been explored in the classification setting prior to this work. We also showed how the trained transformer can exhibit benign overfitting in-context, i.e. in its forward pass the transformer can memorize noisy examples yet still achieve near-optimal test error.

There are a number of natural directions for future research. We relied upon a convex linear transformer architecture which allows for us to identify the pre-trained transformer as the global max-margin solution in parameter space. More general linear transformer architectures are not convex but are often homogeneous in their parameters. In this setting we thus know (Lyu & Li, 2020; Ji & Telgarsky, 2020) that GD has an implicit bias towards first-order stationary (KKT) points of the max-margin problem in parameter space. It may be possible to analzye the consequences of the KKT conditions to develop generalization guarantees (Safran et al., 2022; Frei et al., 2023b;a). For softmax-based transformer architectures, it would be interesting to see if prior works on implicit regularization of GD over these architectures (Ataee Tarzanagh et al., 2023; Tarzanagh et al., 2023; Thrampoulidis, 2024; Vasudeva et al., 2024) can be used to understand in-context learning.

Finally, we assumed that the signal-to-noise ratio in the pre-training data was quite large (see (A1)), and that the pre-training data did not have noisy labels. It would be interesting to understand if pre-training on more difficult or noisy data would result in a qualitatively different algorithm implemented by the pre-trained transformer. We believe the theoretical analysis of pre-training with noisy labels would be significantly different: if we aimed to use the implicit regularization approach based on analyzing solutions to our equation (3), in this setting there may not be a $U$ which satisfies the constraints of (3), and even if such a $U$ does exist then we would be investigating the behavior of a pre-trained transformer which has memorized noisy labels. In this setting it is not clear that generalization is possible.

## ACKNOWLEDGEMENTS

Part of this work was completed while SF was a visitor at the Modern Paradigms in Generalization Program at the Simons Institute for the Theory of Computing. GV is supported by a research grant from Mortimer Zuckerman, the Zuckerman STEM Leadership Program, and by research grants from the Center for New Scientists at the Weizmann Institute of Science, and the Shimon and Golde Picker – Weizmann Annual Grant.

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

Table 1: Notation used throughout the paper.

| Symbol | Description |
|---|---|
| $\mu$ | Cluster mean, $\mu \in \mathbb{R}^d$ |
| $x$ | Features, $x \in \mathbb{R}^d$, $x_{\tau,i} = y_{\tau,i}\mu_\tau + z_{\tau,i}$ |
| $y$ | Labels, $y \in \{\pm 1\}$ |
| $z$ | Noise variables, $z \in \mathbb{R}^d$ |
| $d$ | Feature dimension |
| $\Lambda$ | Cluster noise covariance matrix |
| $\delta$ | Probability of failure |
| $R$ | Norm of cluster means during pre-training |
| $\tilde{R}$ | Norm of cluster means at test time |
| $B$ | Number of pre-training tasks |
| $c_B$ | Quantity such that $B \geq c_B d$ |
| $\rho$ | $(c_B \wedge 1)/(\log^2(2B^2/\delta))$, appears in generalization bounds |
| $N$ | Number of samples per pre-training task |
| $M$ | Number of samples per test-time task |
| $p$ | Label noise flipping rate |
| $E$ | Data matrix $E = \begin{pmatrix} x_1 & x_2 & \cdots & x_N & x_{N+1} \\ y_1 & y_2 & \cdots & y_N & 0 \end{pmatrix} \in \mathbb{R}^{(d+1)\times(N+1)}$ |
| $\widehat{\mu}$ | Mean predictor: $\frac{1}{M}\sum_{i=1}^M y_i x_i$ |
| $\widehat{y}(E(x); W)$ | Neural net output: $\frac{1}{M}\sum_{i=1}^M y_i x_i^T W x = \widehat{\mu}^T W x$ |
| $\widehat{L}(W)$ | Logistic loss |

## A  PROPERTIES OF THE PRE-TRAINING DATASETS

We begin by developing guarantees for various properties of the pre-training datasets, which will form the basis for understanding properties of the max-margin solution (3).

**Lemma A.1.** *There is an absolute constant $c_0 > 1$ such that with probability at least $1 - 10\delta$ over the draws of $\{\mu_\tau, (x_\tau, y_\tau), (x_{\tau,i}, y_{\tau,i})_{i=1}^N\}_{\tau=1}^B$, for all $\tau \in [B]$ and $q \neq \tau$,*

$$\left| \|\widehat{\mu}_\tau\|^2 - R^2 \right| \leq \frac{c_0 R\sqrt{\operatorname{tr}(\Lambda)}\log(2B/\delta)}{\sqrt{Nd}} + 4\frac{\operatorname{tr}(\Lambda) \vee c_0\|\Lambda\|_2 \log(2B/\delta)}{N},$$

$$\left| \|x_\tau\|^2 - R^2 \right| \leq \frac{2c_0 R\sqrt{\operatorname{tr}(\Lambda)}\log(2B/\delta)}{\sqrt{d}} + 4\left(\operatorname{tr}(\Lambda) \vee c_0\|\Lambda\|_2 \log(2B/\delta)\right),$$

$$\left| \langle \widehat{\mu}_q, \widehat{\mu}_\tau \rangle \right| \leq c_0 \left( \frac{R^2}{\sqrt{d}} + \frac{R\sqrt{\operatorname{tr}(\Lambda)}}{\sqrt{Nd}} + \frac{\sqrt{\operatorname{tr}(\Lambda^2)}}{N} \right) \log(2B^2/\delta),$$

$$\left| \langle x_\tau, x_q \rangle \right| \leq c_0 \left( \frac{R^2}{\sqrt{d}} + \frac{R\sqrt{\operatorname{tr}(\Lambda)}}{\sqrt{d}} + \sqrt{\operatorname{tr}(\Lambda^2)} \right) \log(2B^2/\delta),$$

$$\left| \langle \widehat{\mu}_\tau, y_\tau x_\tau \rangle - R^2 \right| \leq c_0 \left( \left[1 + \frac{1}{\sqrt{N}}\right] \frac{R\sqrt{\operatorname{tr}(\Lambda)}}{\sqrt{d}} + \frac{\sqrt{\operatorname{tr}(\Lambda^2)}}{\sqrt{N}} \right) \log(2B/\delta)$$

*Proof.* By definition of $\widehat{\mu}_\tau$ and properties of the Gaussian distribution, there is $z'_\tau \sim \mathsf{N}(0, \Lambda)$ such that

$$\widehat{\mu}_\tau = \frac{1}{N}\sum_{i=1}^N y_{\tau,i}(y_{\tau,i}\mu_\tau + z_{\tau,i}) = \mu_\tau + \frac{1}{N}\sum_{i=1}^N y_{\tau,i}z_{\tau,i} = \mu_\tau + \frac{1}{\sqrt{N}}z'_\tau.$$

Thus for $\tau \neq q$ there are $z'_\tau, z'_q \overset{\text{i.i.d.}}{\sim} \mathsf{N}(0, \Lambda)$ such that

$$\langle \widehat{\mu}_q, \widehat{\mu}_\tau \rangle \overset{\mathrm{d}}{=} \langle \mu_\tau + N^{-1/2}z'_\tau, \mu_q + N^{-1/2}z'_q \rangle$$
$$= \langle \mu_\tau, \mu_q \rangle + N^{-1/2}\langle z'_\tau, \mu_q \rangle + N^{-1/2}\langle z'_q, \mu_\tau \rangle + N^{-1}\langle z'_\tau, z'_q \rangle. \tag{8}$$

We first derive an upper bound for this quantity when $q \neq \tau$. Now, since $\mu_q, \mu_\tau$ are independent and sub-Gaussian random vectors with sub-Gaussian norm at most $cR/\sqrt{d}$ (Vershynin, 2018, Theorem 3.4.6) for some absolute constant $c > 0$, by Vershynin (2018, Lemma 6.2.3), we have for some $c' > 0$ it holds that for any $\beta \in \mathbb{R}$, if $g, g' \overset{\text{i.i.d.}}{\sim} \mathsf{N}(0, I_d)$,

$$\mathbb{E}[\exp(\beta \mu_q^\top \mu_\tau)] \leq \mathbb{E}[\exp(c' R^2 d^{-1} \beta g^\top g')].$$

By Vershynin (2018, Lemma 6.2.2), for some $c_1 > 0$, provided $c|\beta|R^2/d \leq c_1$, it holds that

$$\mathbb{E}[\exp(cR^2 d^{-1} \beta g^\top g')] \leq \exp(c_1 \beta^2 R^4 d^{-2} \|I_d\|_F^2) = \exp(c_1 \beta^2 R^4 d^{-1}).$$

Since $\mu_q, \mu_\tau$ are mean-zero, by Vershynin (2018, Proposition 2.7.1) this implies the quantity $\mu_q^\top \mu_\tau$ is sub-exponential with $\|\mu_q^\top \mu_\tau\|_{\psi_1} \leq c_2 R^2/\sqrt{d}$ for some absolute constant $c_2 > 0$. We therefore have, for some absolute $c_3 > 0$, w.p. at least $1 - \delta$, for all $\tau \in [B]$,

$$|\langle \mu_\tau, \mu_q \rangle| \leq c_3 R^2 d^{-1/2} \log(2B/\delta). \tag{9}$$

Again using Lemmas 6.2.2 and 6.2.3 from (Vershynin, 2018), since $\mu_q$ has sub-Gaussian norm at most $cR/\sqrt{d}$ and $z_\tau' = \Lambda^{1/2} g''$ where $g''$ has sub-Gaussian norm at most $c$, we have

$$\mathbb{E}[\exp(\beta \mu_q^\top z_\tau')] \leq \mathbb{E}[\exp(cRd^{-1/2} \beta g^\top \Lambda^{1/2} g')],$$

and thus provided $cRd^{-1/2}|\beta| \leq c_1/\|\Lambda^{1/2}\|_2$ we have

$$\mathbb{E}[\exp(cRd^{-1/2} \beta g^\top \Lambda^{1/2} g')] \leq \exp(c' R^2 d^{-1} \beta^2 \|\Lambda^{1/2}\|_F^2) = \exp(c' R^2 d^{-1} \beta^2 \operatorname{tr}(\Lambda)).$$

In particular, the quantity $\mu_q^\top z_\tau'$ is sub-exponential with sub-exponential norm $\|\mu_q^\top z_\tau'\|_{\psi_1} \leq c_4 Rd^{-1/2}\sqrt{\operatorname{tr}(\Lambda)}$, and so for some absolute $c_5 > 0$ we have with probability at least $1 - \delta$, for all $q, \tau \in [B]$ with $q \neq \tau$,

$$|\langle \mu_q, z_\tau' \rangle| \leq c_5 Rd^{-1/2}\sqrt{\operatorname{tr}(\Lambda)} \log(2B^2/\delta). \tag{10}$$

For $\langle z_q', z_\tau' \rangle$ with $\tau \neq q$ we can directly use the MGF of Gaussian chaos (Vershynin, 2018, Lemma 6.2.2): $\langle z_q', z_\tau' \rangle = g^\top \Lambda g'$ for i.i.d. $g, g' \sim \mathsf{N}(0, I_d)$ so that for $\beta \leq c/\|\Lambda\|_2$,

$$\mathbb{E}[\exp(\beta \langle z_q', z_\tau' \rangle)] \leq \exp(c_6 \beta^2 \|\Lambda\|_F^2) = \exp(c_6 \beta^2 \operatorname{tr}(\Lambda^2)).$$

In particular, $\|\langle z_\tau', z_q' \rangle\|_{\psi_1} \leq c_7 \sqrt{\operatorname{tr}(\Lambda^2)}$ so that sub-exponential concentration implies that with probability at least $1 - \delta$, for any $q, \tau \in [B]$ with $q \neq \tau$,

$$|\langle z_\tau', z_q' \rangle| \leq c_7 \sqrt{\operatorname{tr}(\Lambda^2)} \log(2B^2/\delta). \tag{11}$$

Putting (9), (10), and (11) into (8) we get for $q \neq \tau$,

$$|\langle \widehat{\mu}_q, \widehat{\mu}_\tau \rangle| = c_8 \left( \frac{R^2}{\sqrt{d}} + \frac{R\sqrt{\operatorname{tr}(\Lambda)}}{\sqrt{Nd}} + \frac{\sqrt{\operatorname{tr}(\Lambda^2)}}{N} \right) \log(2B^2/\delta). \tag{12}$$

As for $\|\widehat{\mu}_\tau\|^2$, from (8) we have

$$\|\widehat{\mu}_\tau\|^2 = \|\mu_\tau\|^2 + 2N^{-1/2} \langle z_\tau', \mu_\tau \rangle + N^{-1} \|z_\tau'\|^2 \tag{13}$$

From here, the same argument used to bound (10) holds since that bound only relied upon the fact that $\mu_q$ and $z_\tau'$ are independent, while $\mu_\tau$ and $z_\tau'$ are independent as well. In particular, with probability at least $1 - \delta$, for all $\tau \in [B]$,

$$|\langle \mu_\tau, z_\tau' \rangle| \leq c_5 Rd^{-1/2}\sqrt{\operatorname{tr}(\Lambda)} \log(2B/\delta). \tag{14}$$

Next, we have that $\|z_\tau'\|^2 \overset{\text{d}}{=} g^\top \Lambda g = \|\Lambda^{1/2} g\|^2$ for $g \sim \mathsf{N}(0, I_d)$. Therefore Vershynin (2018, Theorem 6.3.2) implies $\|\|z_\tau'\|_2 - \|\Lambda^{1/2}\|_F\|_{\psi_2} \leq c\|\Lambda^{1/2}\|_2$ for some absolute constant $c > 0$. Note

that $\|\Lambda^{1/2}\|_F = \sqrt{\text{tr}(\Lambda)}$. And thus by sub-Gaussian concentration, we have for some constant $c_9 > 0$, with probability at least $1 - \delta$, for all $\tau \in [B]$,

$$\left| \|z'_\tau\|_2 - \sqrt{\text{tr}(\Lambda)} \right| \le c_9^{1/2} \|\Lambda^{1/2}\|_2 \sqrt{\log(2B/\delta)} \implies \|z'_\tau\| \le 2(\sqrt{\text{tr}(\Lambda)} \vee c_9^{1/2} \|\Lambda^{1/2}\|_2 \sqrt{\log(2B/\delta)}).$$

In particular,

$$\|z'_\tau\|^2 \le 4 \left( \text{tr}(\Lambda) \vee c_9 \|\Lambda\|_2 \log(2B/\delta) \right). \tag{15}$$

Putting (15) and (14) into (13) and using that $\|\mu_\tau\|^2 = R^2$, we get with probability at least $1 - 2\delta$,

$$\left| \|\widehat{\mu}_\tau\|^2 - R^2 \right| \le \frac{c_5 R \sqrt{\text{tr}(\Lambda)} \log(2B/\delta)}{\sqrt{Nd}} + 4 \frac{\text{tr}(\Lambda) \vee c_9 \|\Lambda\|_2 \log(2B/\delta)}{N}.$$

As for $\|x_\tau\|^2$, by definition,

$$\|x_\tau\|^2 = \|\mu_\tau\|^2 + 2\langle \mu_\tau, z_\tau \rangle + \|z_\tau\|^2 = R^2 + 2\langle \mu_\tau, z_\tau \rangle + \|z_\tau\|^2.$$

Since $z_\tau \sim \mathsf{N}(0, \Lambda)$ has the same distribution as $z'_\tau$, the same analysis used to prove (14) and (15) yields that with probability at least $1 - 2\delta$, for all $\tau \in [B]$,

$$|\langle \mu_\tau, z_\tau \rangle| \le c_5 R d^{-1/2} \sqrt{\text{tr}(\Lambda)} \log(2B/\delta),$$
$$\|z_\tau\|^2 \le 4(\text{tr}(\Lambda) \vee c_9 \|\Lambda\|_2 \log(2B/\delta)).$$

Substituting these into the preceding display we get

$$\left| \|x_\tau\|^2 - R^2 \right| \le \frac{2c_5 R \sqrt{\text{tr}(\Lambda)} \log(2B/\delta)}{\sqrt{d}} + 4 \left( \text{tr}(\Lambda) \vee c_9 \|\Lambda\|_2 \log(2B/\delta) \right)$$

Thus provided $R$ is sufficiently large, then we also have $\|x_\tau\|^2 = \Theta(R^2)$.

Next we bound $|\langle x_\tau, x_q \rangle|$: There are $z'_\tau, z'_q \overset{\text{i.i.d.}}{\sim} \mathsf{N}(0, \Lambda)$ such that

$$\langle y_\tau x_\tau, y_q x_q \rangle \overset{\text{d}}{=} \langle \mu_\tau + z'_\tau, \mu_q + z'_q \rangle.$$

It is clear that the same exact analysis we used to analyze (8) leads to the claim that with probability at least $1 - \delta$, for all $q \ne \tau$:

$$|\langle x_q, x_\tau \rangle| \le c_9 \left( \frac{R^2}{\sqrt{d}} + \frac{R\sqrt{\text{tr}(\Lambda)}}{\sqrt{d}} + \sqrt{\text{tr}(\Lambda^2)} \right) \log(2B^2/\delta). \tag{16}$$

Finally, we consider $y_\tau \widehat{\mu}_\tau^\top x_\tau$. Just as in the previous analyses, there are $z_\tau, z'_\tau \sim \mathsf{N}(0, \Lambda)$ such that

$$\langle \widehat{\mu}_\tau, y_\tau x_\tau \rangle \overset{\text{d}}{=} \langle \mu_\tau + N^{-1/2} z_\tau, \mu_\tau + z'_\tau \rangle$$
$$= \|\mu_\tau\|^2 + N^{-1/2} \langle z_\tau, \mu_\tau \rangle + \langle z'_\tau, \mu_\tau \rangle + N^{-1/2} \langle z_\tau, z'_\tau \rangle.$$

Again using an analysis similar to that used for (8) yields that with probability at least $1 - \delta$, for all $\tau \in [B]$,

$$\left| \langle \widehat{\mu}_\tau, y_\tau x_\tau \rangle - R^2 \right| \le c_{10} \left( \left[ 1 + \frac{1}{\sqrt{N}} \right] \frac{R\sqrt{\text{tr}(\Lambda)}}{\sqrt{d}} + \frac{\sqrt{\text{tr}(\Lambda^2)}}{\sqrt{N}} \right) \log(2B/\delta). \tag{17}$$

Taking a union bound over each of the events shows that all of the desired claims of Lemma A.1 hold with probability at least $1 - 10\delta$.

$\square$

The events of Lemma A.1 hold with probability at least $1 - 10\delta$, independently of the assumptions (A1), (A2), and (A3).[2] Our results will require this event to hold, and so we introduce the following to allow for us to refer to this in later lemmas.

**Definition A.2.** *Let us say that a good run holds if the events of Lemma A.1 hold.*

---

[2] Note, however, that the quantities appearing on the right-hand sides of each inequality in the lemma are only small when these assumptions hold; this is the reason for these assumptions.

# B  ANALYSIS OF MAX-MARGIN SOLUTION

In this section we derive a number of properties of the max-margin solution (3).

**Lemma B.1.** *On a good run, for any $c_B > 0$ and for $C > 1$ sufficiently large under Assumptions (A1) and (A2), the max-margin solution $W$ of Problem 3,*

$$W = \sum_{\tau=1}^{B} \lambda_\tau y_\tau \widehat{\mu}_\tau x_\tau^\top,$$

*is such that the $\lambda_\tau \geq 0$ satisfy the following:*

$$\frac{(c_B \wedge 1)d}{8c_0^2 R^4 \log^2(2B^2/\delta)} \leq \sum_{\tau=1}^{B} \lambda_\tau \leq \frac{4d}{R^4},$$

*where $c_0 > 1$ is the constant from Lemma A.1. Further, we have the inequalities*

$$\frac{(c_B \wedge 1)\sqrt{d}}{16c_0^2 R^2 \log^2(2B^2/\delta)} \leq \|W\|_F \leq \frac{2\sqrt{d}}{R^2},$$

*and*

$$\frac{(c_B \wedge 1)d}{16c_0^2 R^2 \log^2(2B^2/\delta)} \leq \operatorname{tr}(W) \leq \frac{6d}{R^2}.$$

*Proof.* We first derive an upper bound on $\|W\|_F$ by showing that the matrix $U = I_d$ satisfies the constraints of the max-margin problem (3).

$$\begin{aligned}
y_\tau \widehat{\mu}_\tau^\top I x_\tau &= \langle \widehat{\mu}_\tau, y_\tau x_\tau \rangle \\
&\overset{(i)}{\geq} R^2 \left( 1 - 2c_0 \left( \frac{\sqrt{\operatorname{tr}(\Lambda)}}{R\sqrt{d}} + \sqrt{\frac{\operatorname{tr}(\Lambda^2)}{NR^4}} \right) \log(2B/\delta) \right) \\
&\overset{(ii)}{\geq} R^2 \left( 1 - \frac{4c_0}{C} \right) \\
&\geq \frac{1}{2} R^2.
\end{aligned} \tag{18}$$

Inequality $(i)$ uses Lemma A.1, while inequality $(ii)$ holds for $C > 8c_0$ sufficiently large via Assumption (A1). Thus the matrix $2I_d/R^2$ separates the training data with margin at least 1 for every sample. Since $W$ is the minimum Frobenius norm matrix which separates all of the training data with margin 1, this implies

$$\|W\|_F \leq \|2I_d/R^2\|_F = \frac{2\sqrt{d}}{R^2}. \tag{19}$$

On the other hand, by the variational definition of the norm, since $\|I_d/\sqrt{d}\|_F = 1$ we know $\|W\|_F \geq \langle W, I_d/\sqrt{d}\rangle$, and hence

$$\begin{aligned}
\|W\|_F &\geq \frac{1}{\sqrt{d}} \langle W, I_d \rangle \\
&= \frac{1}{\sqrt{d}} \left\langle \sum_{\tau=1}^{B} \lambda_\tau \widehat{\mu}_\tau y_\tau x_\tau^\top, I_d \right\rangle \\
&= \frac{1}{\sqrt{d}} \sum_{\tau=1}^{B} \lambda_\tau \langle \widehat{\mu}_\tau, y_\tau x_\tau \rangle \\
&\geq \frac{1}{\sqrt{d}} \cdot \frac{1}{2} R^2 \sum_{\tau=1}^{B} \lambda_\tau,
\end{aligned} \tag{20}$$

where the last line uses (18). Putting this and the preceding display together, we get

$$\sum_{\tau=1}^{B} \lambda_\tau \leq \frac{4d}{R^4}. \tag{21}$$

Next, we note that by the feasibility conditions of the max-margin problem, we have for any $\tau$,

$$1 \leq y_\tau \widehat{\mu}_\tau^\top W x_\tau$$

$$= \widehat{\mu}_\tau^\top \left( \sum_{q=1}^{B} \lambda_q y_q \widehat{\mu}_q x_q^\top \right) y_\tau x_\tau$$

$$= \sum_{q=1}^{B} \lambda_q \langle \widehat{\mu}_\tau, \widehat{\mu}_q \rangle \langle y_\tau x_\tau, y_q x_q \rangle$$

$$= \lambda_\tau \|\widehat{\mu}_\tau\|^2 \|x_\tau\|^2 + \sum_{q:\, q \neq \tau} \lambda_q \langle \widehat{\mu}_\tau, \widehat{\mu}_q \rangle \langle y_q x_q, x_\tau y_\tau \rangle. \tag{22}$$

Now by Lemma A.1, we have

$$\left| \frac{\|\widehat{\mu}_\tau\|^2}{R^2} - 1 \right| \leq \frac{c_0 \sqrt{\operatorname{tr}(\Lambda) \log(2B/\delta)}}{R\sqrt{Nd}} + 4 \frac{\operatorname{tr}(\Lambda) \vee c_0 \|\Lambda\|_2 \log(2B/\delta)}{NR^2},$$

and

$$\left| \frac{\|x_\tau\|^2}{R^2} - 1 \right| \leq \frac{c_0 \sqrt{\operatorname{tr}(\Lambda) \log(2B/\delta)}}{R\sqrt{d}} + 4 \frac{\operatorname{tr}(\Lambda) \vee c_0 \|\Lambda\|_2 \log(2B/\delta)}{R^2}.$$

Using Assumption (A1) we see that for $C$ sufficiently large we have

$$\|\widehat{\mu}_\tau\|^2 \|x_\tau\|^2 \leq R^4 \left( 1 + \frac{c_0}{NC} + \frac{4c_0}{NC} \right) \cdot \left( 1 + \frac{c_0}{C} + \frac{4c_0}{C} \right) \leq \frac{3}{2} R^4, \tag{23}$$

where the inequality uses Assumption (A1). Likewise, we have

$$\|\widehat{\mu}_\tau\|^2 \|x_\tau\|^2 \geq \frac{1}{2} R^4. \tag{24}$$

As for the cross terms, again using Lemma A.1, for all $\tau \neq q$,

$$|\langle \widehat{\mu}_\tau, \widehat{\mu}_q \rangle| \cdot |\langle x_\tau, x_q \rangle|$$

$$\leq c_0^2 \left( \frac{R^2}{\sqrt{d}} + \frac{R\sqrt{\operatorname{tr}(\Lambda)}}{\sqrt{Nd}} + \frac{\sqrt{\operatorname{tr}(\Lambda^2)}}{N} \right) \cdot \left( \frac{R^2}{\sqrt{d}} + \frac{R\sqrt{\operatorname{tr}(\Lambda)}}{\sqrt{d}} + \sqrt{\operatorname{tr}(\Lambda^2)} \right) \log^2(2B^2/\delta)$$

$$= \frac{c_0^2 R^4}{d} \left( 1 + \frac{\sqrt{\operatorname{tr}(\Lambda)}}{R\sqrt{N}} + \frac{\sqrt{d \operatorname{tr}(\Lambda^2)}}{R^2 N} \right) \cdot \left( 1 + \frac{\sqrt{\operatorname{tr}(\Lambda)}}{R} + \frac{\sqrt{d \operatorname{tr}(\Lambda^2)}}{R^2} \right) \log^2(2B^2/\delta)$$

$$\overset{(i)}{\leq} \frac{c_0^2 R^4}{d} \left( 1 + \frac{1}{C\sqrt{N}} + \frac{1}{4N} \right) \cdot \left( 1 + \frac{1}{C} + \frac{1}{4} \right) \log^2(2B^2/\delta)$$

$$\overset{(ii)}{\leq} \frac{2c_0^2 R^4 \log^2(2B^2/\delta)}{d}. \tag{25}$$

Inequality $(i)$ uses Lemma A.1 and $(ii)$ uses that $C$ is large enough. Putting this together with (23) and substituting these into the consequences of the feasibility condition (22) we get for any $\tau$,

$$1 \leq \frac{3}{2} R^4 \lambda_\tau + \frac{2c_0^2 R^4 \log^2(2B^2/\delta)}{d} \sum_{q:\, q \neq \tau} \lambda_q. \tag{26}$$

We now show that this implies $\sum_{q=1}^{B} \lambda_q \geq \frac{(c_B \wedge 1)d}{8c_0^2 R^4 \log^2(2B^2/\delta)}$. Towards this end, let us first consider the case that there exists some $\tau$ with $\lambda_\tau \leq \frac{c_B d}{4R^4 B}$. Then the preceding display implies

$$1 \leq \frac{3}{2} R^4 \cdot \frac{c_B d}{4R^4 B} + \frac{2c_0^2 R^4 \log^2(2B^2/\delta)}{d} \sum_{q:\, q \neq \tau} \lambda_q$$

$$\leq \frac{3}{4} + \frac{2c_0^2 R^4 \log^2(2B^2/\delta)}{d} \sum_{q:\, q \neq \tau} \lambda_q.$$

The final inequality uses Assumption (A2), i.e. $B \geq c_B d$. Rearranging we see that

$$\sum_{q=1}^{B} \lambda_q \geq \sum_{q:\, q \neq \tau} \lambda_q \geq \frac{d}{8c_0^2 R^4 \log^2(2B^2/\delta)} \geq \frac{(c_B \wedge 1)d}{8c_0^2 R^4 \log^2(2B^2/\delta)}.$$

Let us now consider the only remaining case, whereby for each $\tau$ we have $\lambda_\tau > \frac{c_B d}{4R^4 B}$. Then,

$$\sum_{q=1}^{B} \lambda_q > \sum_{q=1}^{B} \frac{c_B d}{4R^4 B} = \frac{c_B d}{4R^4} \geq \frac{(c_B \wedge 1)d}{8c_0^2 R^4 \log^2(2B^2/\delta)},$$

where the final inequality uses that $c_0 \geq 1$. We therefore have $\sum_{q=1}^{B} \lambda_q \geq \frac{(c_B \wedge 1)d}{8c_0^2 R^4 \log^2(2B^2/\delta)}$, which together with (21) completes the proof for the upper and lower bounds of $\sum_{q=1}^{B} \lambda_q$. The upper bound for the Frobenius norm of $W$ follow by (19), while the lower bound follows by $\sum_{q=1}^{B} \lambda_q \geq \frac{(c_B \wedge 1)d}{8c_0^2 R^4 \log^2(2B^2/\delta)}$ and (20).

For the trace term, we have,

$$\mathrm{tr}(W_{\mathsf{MM}}) = \mathrm{tr}\left( \sum_{\tau=1}^{B} \lambda_\tau y_\tau \widehat{\mu}_\tau x_\tau^\top \right)$$

$$= \sum_{\tau=1}^{B} \lambda_\tau \, \mathrm{tr}(y_\tau \widehat{\mu}_\tau x_\tau^\top)$$

$$= \sum_{\tau=1}^{B} \lambda_\tau y_\tau x_\tau^\top \widehat{\mu}_\tau$$

$$\overset{(i)}{\geq} \sum_{\tau=1}^{B} \lambda_\tau R^2 / 2$$

$$\overset{(ii)}{\geq} \frac{(c_B \wedge 1)d}{16c_0^2 R^2 \log^2(2B^2/\delta)} \tag{27}$$

Inequality $(i)$ uses (18) while inequality $(ii)$ uses the lower bound for $\sum_\tau \lambda_\tau$. Similarly, by Lemma A.1 we have

$$|\langle \widehat{\mu}_\tau, y_\tau x_\tau \rangle| \leq R^2 \left( 1 + c_0 \left( \left[1 + \frac{1}{\sqrt{N}}\right] \frac{\sqrt{\mathrm{tr}(\Lambda)}}{R\sqrt{d}} + \frac{\sqrt{\mathrm{tr}(\Lambda^2)}}{\sqrt{R^4 N}} \right) \log(2B/\delta) \right) \leq \frac{3}{2} R^2,$$

where the last inequality uses Assumption (A1). Therefore,

$$\mathrm{tr}(W_{\mathsf{MM}}) = \sum_{\tau=1}^{B} \lambda_\tau y_\tau x_\tau^\top \widehat{\mu}_\tau$$

$$\leq \sum_{\tau=1}^{B} \lambda_\tau \cdot \frac{3}{2} R^2$$

$$\leq \frac{6d}{R^2},$$

where the last inequality uses the upper bound for $\sum_{\tau=1}^{B} \lambda_\tau$ proved earlier. $\qquad\square$

## C  Proof of Theorem 4.1

Our goal is to bound

$$R(W_{\mathsf{MM}}) = \mathbb{P}_{(x_i, y_i)_1^{M+1},\, \mu}(\mathrm{sign}(\widehat{y}(E; W_{\mathsf{MM}})) \neq y_{M+1})$$

$$= \mathbb{P}\left( \left[ \frac{1}{M} \sum_{i=1}^{M} y_i x_i \right]^\top W_{\mathsf{MM}} y_{M+1} x_{M+1} < 0 \right). \tag{28}$$

The probability above is over the draws of $\mu$ and of $(x_i, y_i)_{i=1}^{M+1}$ and $\mu$. The matrix $W_{\mathsf{MM}}$ is the max-margin solution when training on the in-context tasks (3). To prove the risk is close to the label-flipping noise rate $p$, it suffices to show that the error on clean examples is small:

$$
\begin{aligned}
R(W_{\mathsf{MM}}) &= \mathbb{P}\left(y_{M+1} \cdot \mathrm{sign}(\widehat{y}(E; W_{\mathsf{MM}})) < 0,\ y_{M+1} \neq \tilde{y}_{M+1}\right) \\
&\quad + \mathbb{P}\left(y_{M+1} \cdot \mathrm{sign}(\widehat{y}(E; W_{\mathsf{MM}})) < 0,\ y_{M+1} = \tilde{y}_{M+1}\right) \\
&\leq p + \mathbb{P}\left(y_{M+1} \cdot \mathrm{sign}(\widehat{y}(E; W_{\mathsf{MM}})) < 0,\ y_{M+1} = \tilde{y}_{M+1}\right).
\end{aligned} \tag{29}
$$

For notational simplicity, let us denote $\widehat{y} := \widehat{y}(E; W_{\mathsf{MM}})$. Let us also denote the event $\tilde{A}$ as

$$
\tilde{A} := \{y_{M+1}\widehat{y} < 0\} = \{y_{M+1}\widehat{y} - \mathbb{E}[y_{M+1}\widehat{y}] < -\mathbb{E}[y_{M+1}\widehat{y}]\},
$$

so that $R(W_{\mathsf{MM}}) = \mathbb{P}(\tilde{A})$. Further, let us introduce the sets $\mathcal{C}, \mathcal{N} \subset [M]$ as the clean and noisy test examples respectively ($|\mathcal{C}| + |\mathcal{N}| = M$) so that

$$
y_i = \tilde{y}_i\ \forall i \in \mathcal{C}, \quad y_i = -\tilde{y}_i\ \forall i \in \mathcal{N}.
$$

Then we have the identities

$$
\begin{aligned}
y_i x_i &= \mu + y_i z_i, \quad i \in \mathcal{C}, \\
y_i x_i &= -\mu + y_i z_i, \quad i \in \mathcal{N}.
\end{aligned}
$$

Therefore there exist independent $\zeta, \zeta_{M+1} \sim \mathsf{N}(0, \Lambda)$ such that

$$
\frac{1}{M}\sum_{i=1}^{M} y_i x_i \stackrel{\mathrm{d}}{=} \left(1 - \frac{2|\mathcal{N}|}{M}\right)\mu + \frac{1}{\sqrt{M}}\zeta, \quad \tilde{y}_{M+1}x_{M+1} \stackrel{\mathrm{d}}{=} \mu + \zeta_{M+1},
$$

where we have used that $|\mathcal{C}| - |\mathcal{N}| = M - 2|\mathcal{N}|$.

In particular, if we define

$$
A := \left\{\left((1 - 2|\mathcal{N}|/M)\mu + M^{-1/2}\zeta\right)^{\top} W(\mu + \zeta_{M+1}) - \mathbb{E}[y_{M+1}\widehat{y}] < -\mathbb{E}[y_{M+1}\widehat{y}]\right\}
$$

then we have

$$
\mathbb{P}\left(y_{M+1}\widehat{y} - \mathbb{E}[y_{M+1}\widehat{y}] < -\mathbb{E}[y_{M+1}\widehat{y}],\ y_{M+1} = \tilde{y}_{M+1}\right) = \mathbb{P}(A).
$$

Continuing from (29) this means for any $\alpha_0, \alpha_1, \alpha_2, \alpha_3 > 0$,

$$
\begin{aligned}
&R(W_{\mathsf{MM}}) \\
&\leq p + \mathbb{P}\left(\left((1 - 2|\mathcal{N}|/M)\mu + M^{-1/2}\zeta\right)^{\top} W(\mu + \zeta_{M+1}) - \mathbb{E}[y_{M+1}\widehat{y}] < -\mathbb{E}[y_{M+1}\widehat{y}]\right) \\
&\leq p + \mathbb{P}(|\mathcal{N}|/M \leq \alpha_0) + \mathbb{P}(|M^{-1/2}\zeta^{\top}W_{\mathsf{MM}}\mu| > \alpha_1) \\
&\quad + \mathbb{P}(|M^{-1/2}\zeta^{\top}W_{\mathsf{MM}}\zeta_{M+1}| > \alpha_2) + \mathbb{P}((1 - 2|\mathcal{N}|/M)|\mu^{\top}W_{\mathsf{MM}}\zeta_{M+1}| > \alpha_3) \\
&\quad + \mathbb{P}\bigg(A \cap \{|\mathcal{N}|/M \leq \alpha_0\} \cap \{|M^{-1/2}\zeta^{\top}W_{\mathsf{MM}}\mu| \leq \alpha_1\} \\
&\qquad\qquad \cap \{|M^{-1/2}\zeta^{\top}W_{\mathsf{MM}}\zeta_{M+1}| \leq \alpha_2\} \cap \{|\mu^{\top}W_{\mathsf{MM}}\zeta_{M+1}| \leq \alpha_3\}\bigg).
\end{aligned} \tag{30}
$$

In particular, to derive an upper bound on the risk it suffices to derive a lower bound on $\mu^{\top}W_{\mathsf{MM}}\mu$ and upper bounds on each of the absolute values of the four quantities $|\mathcal{N}|/M$, $\zeta^{\top}W_{\mathsf{MM}}\mu, \zeta^{\top}W_{\mathsf{MM}}\zeta_{M+1}$ and $\mu^{\top}W\zeta_{N+1}$. We shall do so in what follows.

## C.1 $|\mathcal{N}|/M$

**Lemma C.1.** *There is some constant $c > 0$ such that for any $t \geq 0$, we have*

$$
\begin{cases}
|\mathcal{N}| = 0, & p = 0, \\
\mathbb{P}\left(\left|\frac{|\mathcal{N}|}{M} - p\right| \geq t\right) \leq 2\exp(-2t^2 M), & p > 0.
\end{cases} \tag{31}
$$

*Proof.* If $p = 0$, there is no label flipping noise and so $|\mathcal{N}| = 0$ deterministically. For the $p > 0$ case, by definition, $|\mathcal{N}| = \sum_{i=1}^{M} \mathbb{1}(y_i \neq \tilde{y}_i)$ is a sum of $M$ independent random variables bounded between 0 and 1 with mean $p$. By Hoeffding's inequality, for any $u \geq 0$ we have

$$\mathbb{P}(|\,|\mathcal{N}| - Mp\,| \geq u) = \mathbb{P}(|\,|\mathcal{N}|/M - p\,| \geq u/M) \leq 2\exp\left(-\frac{2u^2}{M}\right).$$

Setting $u = Mt$ completes the proof. $\qquad\square$

## C.2 $\mu^\top W \mu$

The quantity $\mu^\top W \mu$ is a quadratic form, and we can provide a concentration inequality for this in the following lemma.

**Lemma C.2** (Hanson-Wright for uniform on the sphere). *Let $\tilde{R} > 0$ and $Q \in \tilde{\mathbb{R}}^{d \times d}$ be a matrix. If $\mu \sim \mathsf{Unif}(\tilde{R} \cdot \mathbb{S}^{d-1})$, then for any $t \geq 0$,*

$$\mathbb{P}\left(\left|\mu^\top Q\mu - \frac{\tilde{R}^2}{d}\operatorname{tr}(Q)\right| \geq t\right) = \mathbb{P}\left(|\mu^\top Q\mu - \mathbb{E}[\mu^\top Q\mu]| \geq t\right) \leq 2\exp\left(-c\min\left(\frac{t^2 d^2}{\tilde{R}^4\|Q\|_F^2}, \frac{td}{\tilde{R}^2\|Q\|_2}\right)\right).$$

Since $\mu^\top Q\mu$ is a quadratic form, we would like to use the Hanson-Wright inequality here. But the trouble is that for $\mu \sim \tilde{R}\mathsf{Unif}(\mathbb{S}^{d-1})$, the components of $\mu$ are *not* independent, so the standard Hanson-Wright inequality is not directly applicable, but requires some additional work. Our proof instead leverages the following:

**Theorem C.3** (Adamczak (2015), Theorem 2.5). *Let $X$ be a mean-zero random vector in $\mathbb{R}^d$. If $X$ satisfies the $K$-convex concentration property, i.e. there exists $K$ such that for every 1-Lipschitz convex function $\phi : \mathbb{R}^d \to \mathbb{R}$ we have $\mathbb{E}|\phi(X)| < \infty$ and for every $t > 0$,*

$$\mathbb{P}(|\phi(X) - \mathbb{E}\phi(X)| \geq t) \leq 2\exp(-t^2/K^2),$$

*then there exists $c > 0$ such that for any matrix $Q \in \mathbb{R}^{d \times d}$ and every $t > 0$,*

$$\mathbb{P}(|X^\top QX - \mathbb{E}[X^\top QX]| \geq t) \leq 2\exp\left(-c\min\left(\frac{t^2}{2K^4\|Q\|_F^2}, \frac{t}{K^2\|Q\|_2}\right)\right).$$

*Proof of Lemma C.2.* By Adamczak (2015), a sufficient condition for a Hanson-Wright-type inequality to hold is that the random vector $\mu$ satisfies Lipschitz concentration. By Vershynin (2018, Theorem 5.1.4) we have that for some constant $c > 0$, for any 1-Lipschitz function $f$ on the sphere $\sqrt{d} \cdot \mathbb{S}^{d-1}$ and for $X \sim \mathsf{Unif}(\sqrt{d}\mathbb{S}^{d-1})$,

$$\mathbb{P}(|f(X) - \mathbb{E}[f(X)]| \geq t) \leq 2\exp(-ct^2).$$

Thus if we consider $g : \tilde{R} \cdot \mathbb{S}^{d-1} \to \mathbb{R}$ which is 1-Lipschitz and is defined on the sphere of radius $\tilde{R}$, then $Y = \frac{\tilde{R}}{\sqrt{d}}X$ where $X$ is uniform on the sphere of radius $\tilde{R}$. In particular, the function $g(Y) := g(\frac{\tilde{R}}{\sqrt{d}}X)$ is the composition of a 1-Lipschitz function and a $\tilde{R}/\sqrt{d}$-Lipschitz function defined on the sphere of radius $\sqrt{d}$, and we therefore see that

$$\mathbb{P}(|g(Y) - \mathbb{E}g(Y)| \geq t) \leq 2\exp(-ct^2 d/\tilde{R}^2).$$

That is, any 1-Lipschitz function on the sphere of radius $\tilde{R}$ satisfies the $K$-convex concentration property with $K = \tilde{R}/\sqrt{d}$. Thus Theorem C.3 implies for any matrix $Q$,

$$\mathbb{P}(|\mu^\top Q\mu - \mathbb{E}[\mu^\top Q\mu]| \geq t) \leq 2\exp\left(-c\min\left(\frac{t^2 d^2}{\tilde{R}^4\|Q\|_F^2}, \frac{td}{\tilde{R}^2\|Q\|_2}\right)\right).$$

Finally, note that by rotational symmetry we have $\mathbb{E}[\mu] = 0$ and $\mathbb{E}[\mu_i^2] = \tilde{R}^2/d$ for each $i \in [d]$. Additionally, for distinct components $i \neq j$, $\mathbb{E}[\mu_i\mu_j] = \mathbb{E}[\mu_i(-\mu_j)] = 0$ and hence $\mathbb{E}[\mu^\top Q\mu] = \sum_{i=1}^{d}\mathbb{E}[\mu_i^2 Q_{ii}] = \frac{\tilde{R}^2}{d}\operatorname{tr}(Q)$. $\qquad\square$

## C.3 $\mu^\top W\zeta$ AND $\mu^\top W\zeta_{M+1}$

**Lemma C.4.** *Let* $\mu \sim \tilde{R} \cdot \mathsf{Unif}(\mathbb{S}^{d-1})$, $g \sim \mathsf{N}(0, I_d)$ *(independent of* $\mu$*) and let* $Q \in \mathbb{R}^{d\times d}$ *be a matrix. There is an absolute constant* $c > 0$ *such that for any* $t \geq 0$,

$$\mathbb{P}\left(|\mu^\top Qg| \geq t\right) \leq 2\exp\left(-\frac{ct\sqrt{d}}{\tilde{R}\|Q\|_F}\right).$$

*Proof.* The uniform distribution on the sphere of radius $\tilde{R}$ is sub-Gaussian with sub-Gaussian norm satisfying $\|\mu\|_{\psi_2} \leq c\tilde{R}/\sqrt{d}$ by Vershynin (2018, Theorem 3.4.6), while $\|g\|_{\psi_2} \leq c$, for some absolute constant $c > 0$. By Vershynin (2018, Lemma 6.2.3), this implies that for independent $g_1, g_2 \sim \mathsf{N}(0, I_d)$ and any $\beta \in \mathbb{R}$,

$$\mathbb{E}\exp\left(\beta\frac{\sqrt{d}}{\tilde{R}}\mu^\top Qg\right) \leq \mathbb{E}\exp\left(c_1\beta g_1^\top Qg_2\right).$$

Then using the moment-generating function of Gaussian chaos (Vershynin, 2018, Lemma 6.2.2), for $\beta$ satisfying $|\beta| \leq c_2/\|Q\|_2$ we have

$$\mathbb{E}\exp\left(\beta\frac{\sqrt{d}}{\tilde{R}}\mu^\top Qg\right) \leq \mathbb{E}\exp\left(c_1\beta g_1^\top Qg_2\right)$$

$$\leq \exp(c_3\beta^2\|Q\|_F^2).$$

That is, the random variable $\tilde{R}^{-1}\sqrt{d}\mu^\top Qg$ is mean-zero and has sub-exponential norm at most $\max(c_2^{-1}, c_3)\|Q\|_F$. There is therefore a constant $c_4 > 0$ such that for any $u \geq 0$,

$$\mathbb{P}(|\tilde{R}^{-1}\sqrt{d}\mu^\top Qg| \geq u) = \mathbb{P}\left(|\mu^\top Qg| \geq \frac{\tilde{R}u}{\sqrt{d}}\right) \leq 2\exp(-cu/\|Q\|_F).$$

Setting $u = t\sqrt{d}/\tilde{R}$ we get

$$\mathbb{P}\left(|\mu^\top Qg| \geq t\right) \leq 2\exp\left(-\frac{ct\sqrt{d}}{\tilde{R}\|Q\|_F}\right)$$

$\square$

## C.4 $\zeta^\top W\zeta_{M+1}$

**Lemma C.5.** *Let* $\zeta, \zeta' \overset{\text{i.i.d.}}{\sim} \mathsf{N}(0, \Lambda)$ *and let* $Q \in \mathbb{R}^{d\times d}$ *be a matrix. There is a constant* $c > 0$ *such that for all* $t \geq 0$,

$$\mathbb{P}(|\zeta^\top W\zeta'| \geq t) \leq 2\exp\left(-\frac{ct}{\|\Lambda^{1/2}Q\Lambda^{1/2}Q\|_F}\right) \leq 2\exp\left(-\frac{ct}{\|\Lambda\|_2\|Q\|_F}\right).$$

*Proof.* We can write $\zeta^\top W\zeta_{M+1}$ as $g_1^\top\Lambda^{1/2}W\Lambda^{1/2}g_2$ where $g_1, g_2 \sim \mathsf{N}(0, I_d)$ are independent, i.e. it is a Gaussian chaos random variable. By Vershynin (2018, Lemma 6.2.2) this random variable is sub-exponential with sub-exponential norm at most $c\|\Lambda^{1/2}W\Lambda^{1/2}\|_F$. Then standard sub-exponential concentration (e.g. Vershynin (2018, Proposition 2.7.1)) completes the proof.

$\square$

## C.5 $\mathbb{E}[\mu^\top W_{\mathsf{MM}}\mu]$

**Lemma C.6.** *On a good run, for any* $c_B > 0$ *and for* $C > 1$ *sufficiently large under assumptions (A1) through (A3), the max-margin solution* $W_{\mathsf{MM}}$ *satisfies*

$$\frac{(c_B \wedge 1)\tilde{R}^2}{16c_0^2R^2\log^2(2B^2/\delta)} \leq \mathbb{E}[\mu^\top W_{\mathsf{MM}}\mu] \leq \frac{6\tilde{R}^2}{R^2}.$$

*Proof.* By definition,

$$\mathbb{E}\left[\mu^\top W_{\mathsf{MM}}\mu\right] = \frac{\tilde{R}^2}{d}\,\mathrm{tr}(W_{\mathsf{MM}}). \tag{32}$$

We therefore need only upper and lower bounds on $\mathrm{tr}(W_{\mathsf{MM}})$. Using the bounds on $\mathrm{tr}(W_{\mathsf{MM}})$ from Lemma B.1, we get the desired inequalities. $\qquad\square$

### C.6 PUTTING IT ALL TOGETHER

Let's denote $R(W_{\mathsf{MM}})$ the test error for a clean example $(y_{M+1} = \tilde{y}_{M+1})$. Returning to the decomposition (30) and using (29), we have

$$R(W_{\mathsf{MM}}) \leq p + \mathbb{P}\Bigg( (1 - 2|\mathcal{N}|/M)\mu^\top W\mu - \mathbb{E}[y\hat{y}]$$
$$< -\mathbb{E}[y\hat{y}] - M^{-1/2}\zeta^\top W\mu - (1 - 2|\mathcal{N}|/M)\mu^\top W\zeta_{M+1} - M^{-1/2}\zeta^\top W\zeta_{M+1}\Bigg) \tag{33}$$

We'll consider two cases, depending upon whether the label-flipping noise rate $p = 0$ or $p > 0$. For notational simplicity let's denote $\rho$ as the quantity

$$\rho := \frac{c_B \wedge 1}{16c_0^2 \log^2(2B^2/\delta)} \in (0,1).$$

Then Lemma C.6 states

$$\rho \cdot \frac{\tilde{R}^2}{R^2} \leq \mathbb{E}[\mu^\top W\mu] \leq \frac{6\tilde{R}^2}{R^2}. \tag{34}$$

**Noiseless case** $(p = 0)$. Since $p = 0$ we know $|\mathcal{N}| = 0$. From (33) we have

$$R(W_{\mathsf{MM}})$$
$$= \mathbb{P}\Bigg( \mu^\top W\mu - \mathbb{E}[\mu^\top W\mu] < -\mathbb{E}[\mu^\top W\mu] - M^{-1/2}\zeta^\top W\mu - \mu^\top W\zeta_{M+1} - M^{-1/2}\zeta^\top W\zeta_{M+1}\Bigg)$$
$$\overset{(i)}{\leq} \mathbb{P}\Bigg( \mu^\top W\mu - \mathbb{E}[\mu^\top W\mu] < -\frac{\rho\tilde{R}^2}{R^2} - M^{-1/2}\zeta^\top W\mu - \mu^\top W\zeta_{M+1} - M^{-1/2}\zeta^\top W\zeta_{M+1}\Bigg)$$
$$\leq \mathbb{P}\Bigg( \mu^\top W\mu - \mathbb{E}[\mu^\top W\mu] < -\frac{\rho\tilde{R}^2}{2R^2}\Bigg) + \mathbb{P}\left( |M^{-1/2}\zeta^\top W\mu| \geq \frac{\rho\tilde{R}^2}{8R^2}\right) + \mathbb{P}\left( |\mu^\top W\zeta_{M+1}| \geq \frac{\rho\tilde{R}^2}{8R^2}\right)$$
$$+ \mathbb{P}\left( \left| M^{-1/2}\zeta^\top W\zeta_{M+1}\right| \geq \frac{\rho\tilde{R}^2}{8R^2}\right). \tag{35}$$

Inequality $(i)$ uses Lemma C.6. We now proceed by bounding each of the remaining terms in the inequality above.

For the first term we can use Lemma C.2 with $t = \rho\tilde{R}^2/2R^2$. To do so we need to examine the quantity $\frac{td}{\tilde{R}^2\|W\|_F}$, which we have

$$\frac{td}{\tilde{R}^2\|W\|_F} = \frac{\rho d}{2R^2\|W\|_F} \geq \frac{\rho d}{2R^2 \cdot 2\sqrt{d}R^{-2}} = \frac{\rho\sqrt{d}}{4}.$$

Let's assume for the moment that $\rho\sqrt{d}/4 > 1$. We can then apply Lemma C.2 (noting that this lemma holds if we replace the $\|Q\|_2$ with $\|Q\|_F$),

$$\mathbb{P}\left(\mu^\top W\mu - \mathbb{E}[\mu^\top W\mu] < -\frac{\rho\tilde{R}^2}{2R^2}\right) \leq 2\exp\left(-c\min\left(\frac{d^2}{\tilde{R}^4\|W\|_F^2}\cdot\frac{\rho^2\tilde{R}^4}{4R^4}, \frac{d}{\tilde{R}^2\|W\|_F}\cdot\frac{\rho\tilde{R}^2}{2R^2}\right)\right)$$
$$\leq 2\exp\left(-\frac{cd\rho}{2\|W\|_F R^2}\right)$$
$$\overset{(i)}{\leq} 2\exp\left(-\frac{c\rho\sqrt{d}}{4}\right). \tag{36}$$

Note that if $\rho\sqrt{d}/4 \leq 1$ then the above bound still holds since $\mathbb{P}(\cdot) \leq 1$ is a trivial inequality. This completes the bound for the first term in (35).

The second and third terms in (35) can be bounded using Lemma C.4: since $\zeta_{M+1} = \Lambda^{1/2}g$ for $g \sim \mathsf{N}(0, I_d)$,

$$\mathbb{P}\left(|\mu^\top W\zeta_{M+1}| \geq \frac{\rho\tilde{R}^2}{8R^2}\right) = \mathbb{P}\left(|\mu^\top W\Lambda^{1/2}g| \geq \frac{\rho\tilde{R}^2}{8R^2}\right)$$
$$\leq 2\exp\left(-\frac{c\sqrt{d}}{\tilde{R}\|W\Lambda^{1/2}\|_F}\cdot\frac{\rho\tilde{R}^2}{8R^2}\right)$$
$$\leq 2\exp\left(-\frac{c\sqrt{d}}{\tilde{R}\|\Lambda^{1/2}\|_2\|W\|_F}\cdot\frac{\rho\tilde{R}^2}{8R^2}\right)$$
$$\overset{(i)}{\leq} 2\exp\left(-\frac{c\rho\tilde{R}\sqrt{d}}{8R^2\|\Lambda^{1/2}\|_2\cdot 2\sqrt{d}R^{-2}}\right)$$
$$= 2\exp\left(-\frac{c\rho\tilde{R}}{16\|\Lambda^{1/2}\|_2}\right). \tag{37}$$

The inequality $(i)$ uses the upper bound $\|W\| \leq 2\sqrt{d}/R^2$ from Lemma B.1.

For the final term in (35) we can use Lemma C.5,

$$\mathbb{P}\left(\left|M^{-1/2}\zeta^\top W\zeta_{M+1}\right| \geq \frac{\rho\tilde{R}^2}{8R^2}\right) = \mathbb{P}\left(|\zeta^\top W\zeta_{M+1}| \geq \frac{\rho M^{1/2}\tilde{R}^2}{8R^2}\right)$$
$$\leq 2\exp\left(-\frac{c}{\|\Lambda\|_2\|W\|_F}\cdot\frac{\rho M^{1/2}\tilde{R}^2}{8R^2}\right)$$
$$\overset{(i)}{\leq} 2\exp\left(-\frac{c}{\|\Lambda\|_2\cdot 2\sqrt{d}R^{-2}}\cdot\frac{\rho M^{1/2}\tilde{R}^2}{8R^2}\right)$$
$$= 2\exp\left(-\frac{c\rho M^{1/2}\tilde{R}^2}{16\|\Lambda\|_2\sqrt{d}}\right). \tag{38}$$

Again $(i)$ uses Lemma C.5.

Putting together (36), (37), (38) we get

$$R(W_{\mathsf{MM}}) \leq 2\exp\left(-\frac{c\rho\sqrt{d}}{4}\right) + 4\exp\left(-\frac{c\rho\tilde{R}}{16\|\Lambda^{1/2}\|_2}\right) + 2\exp\left(-\frac{c\rho M^{1/2}\tilde{R}^2}{16\|\Lambda\|_2\sqrt{d}}\right).$$

**Noisy case.** Returning to (33): as before, the 'signal' in the problem comes from the term $(1 - 2|\mathcal{N}|/M)\mu^\top W\mu$, which ideally is large and positive. It is natural that we should require more samples $M$ the closer the noise rate $p$ gets to $1/2$ (namely, the smaller $c_p$ is), since otherwise with

nontrivial probability we will see more noisy examples than clean ones and learning should be impossible. To this end, let's assume that $p \leq \frac{1}{2} - c_p$ for some absolute constant $c_p \in (0, 1/2)$. Continuing from (31) which shows that

$$\mathbb{P}\left(1 - 2|\mathcal{N}|/M > \frac{1}{2}c_p\right) = \mathbb{P}\left(\frac{2|\mathcal{N}|}{M} - 1 + 2c_p \leq \frac{6}{4}c_p\right) \leq 2\exp\left(-\frac{18c_p^2 M}{16}\right) \leq 2\exp(-c_p^2 M). \tag{39}$$

Let's call the event

$$\mathcal{E} := \left\{1 - \frac{2|\mathcal{N}|}{M} > \frac{1}{2}c_p\right\}.$$

Continuing from (33), since $1 - 2|\mathcal{N}|/M > 0$ on $\mathcal{E}$ we have

$$
\begin{aligned}
R(W_{\mathsf{MM}}) &\leq p \\
&+ \mathbb{P}\left(\mu^\top W\mu - \frac{\mathbb{E}[y\widehat{y}]}{(1 - 2|\mathcal{N}|/M)} < -\frac{\mathbb{E}[y\widehat{y}] - M^{-1/2}\zeta^\top W\mu - (1 - 2|\mathcal{N}|/M)\mu^\top W\zeta_{M+1} - M^{-1/2}\zeta^\top W\zeta_{M+1}}{(1 - 2|\mathcal{N}|/M)}\right) \\
&\leq p + \mathbb{P}(\mathcal{E}^c) \\
&+ \mathbb{P}\left(\mathcal{E}, \mu^\top W\mu - \frac{\mathbb{E}[y\widehat{y}]}{(1 - 2|\mathcal{N}|/M)} < -\frac{\mathbb{E}[y\widehat{y}] - M^{-1/2}\zeta^\top W\mu - (1 - 2|\mathcal{N}|/M)\mu^\top W\zeta_{M+1} - M^{-1/2}\zeta^\top W\zeta_{M+1}}{(1 - 2|\mathcal{N}|/M)}\right) \\
&\overset{(i)}{\leq} p + 2\exp(-c_p^2 M) \\
&+ \mathbb{P}\left(\mathcal{E}, \mu^\top W\mu - \frac{\mathbb{E}[y\widehat{y}]}{(1 - 2|\mathcal{N}|/M)} < -\frac{\mathbb{E}[y\widehat{y}] - M^{-1/2}\zeta^\top W\mu - (1 - 2|\mathcal{N}|/M)\mu^\top W\zeta_{M+1} - M^{-1/2}\zeta^\top W\zeta_{M+1}}{(1 - 2|\mathcal{N}|/M)}\right) \\
&\leq p + 2\exp(-c_p^2 M) \\
&+ \mathbb{P}\left(\mathcal{E}, \mu^\top W\mu - \frac{\mathbb{E}[y\widehat{y}]}{(1 - 2|\mathcal{N}|/M)}\right. \\
&\qquad\qquad\left. < \frac{-\mathbb{E}[y\widehat{y}]}{1 - 2|\mathcal{N}|/M} + 2c_p^{-1}\left[M^{-1/2}|\zeta^\top W\mu| + |1 - 2|\mathcal{N}|/M| \cdot |\mu^\top W\zeta_{M+1}| + |M^{-1/2}\zeta^\top W\zeta_{M+1}|\right]\right).
\end{aligned}
\tag{40}
$$

The inequality $(i)$ uses (39). We want to deal with the quantity $\mu^\top W\mu - \frac{\mathbb{E}[\mu^\top W\mu]}{1 - 2|\mathcal{N}|/M} = \frac{\mathbb{E}[y\widehat{y}]}{(1 - 2|\mathcal{N}|/M)}$ on the event $\mathcal{E}$. We have,

$$
\begin{aligned}
\mu^\top W\mu - \frac{\tilde{R}^2(1 - 2p)\operatorname{tr}(W)}{d(1 - 2|\mathcal{N}|/M)} &= \mu^\top W\mu - \frac{\tilde{R}^2\operatorname{tr}(W)}{d} \cdot \frac{1 - 2p}{1 - 2|\mathcal{N}|/m} \\
&= \mu^\top W\mu - \frac{\tilde{R}^2\operatorname{tr}(W)}{d} + \frac{\tilde{R}^2\operatorname{tr}(W)}{d} \cdot \left(1 - \frac{1 - 2p}{1 - 2|\mathcal{N}|/m}\right) \\
&= \mu^\top W\mu - \frac{\tilde{R}^2\operatorname{tr}(W)}{d} - \frac{\tilde{R}^2\operatorname{tr}(W)}{d} \cdot \frac{-2p + 2|\mathcal{N}|/M}{1 - 2|\mathcal{N}|/M}.
\end{aligned}
$$

Continuing from (40) this means

$$R(W_{\mathsf{MM}}) \le p + 2\exp(-c_p^2 M)$$

$$+ \mathbb{P}\Bigg(\mathcal{E}, \ \mu^\top W \mu - \frac{\tilde{R}^2 \operatorname{tr}(W)}{d} - \frac{\tilde{R}^2 \operatorname{tr}(W)}{d} \cdot \frac{-2p + 2|\mathcal{N}|/M}{1 - 2|\mathcal{N}|/M}$$

$$< \frac{-\tilde{R}^2(1 - 2p)\operatorname{tr}(W)}{d(1 - 2|\mathcal{N}|/M)} + 2c_p^{-1}\left[M^{-1/2}|\zeta^\top W \mu| + |1 - 2|\mathcal{N}|/M| \cdot |\mu^\top W \zeta_{M+1}| + |M^{-1/2}\zeta^\top W \zeta_{M+1}|\right]\Bigg)$$

$$= p + 2\exp(-c_p^2 M)$$

$$+ \mathbb{P}\Bigg(\mathcal{E}, \ \mu^\top W \mu - \mathbb{E}[\mu^\top W \mu]$$

$$< \frac{-\tilde{R}^2 \operatorname{tr}(W)}{d} + 2c_p^{-1}\left[M^{-1/2}|\zeta^\top W \mu| + |1 - 2|\mathcal{N}|/M| \cdot |\mu^\top W \zeta_{M+1}| + |M^{-1/2}\zeta^\top W \zeta_{M+1}|\right]\Bigg)$$

$$\overset{(i)}{\le} p + 2\exp(-c_p^2 M)$$

$$+ \mathbb{P}\Bigg(\mathcal{E}, \ \mu^\top W \mu - \mathbb{E}[\mu^\top W \mu]$$

$$< -\rho \cdot \frac{\tilde{R}^2}{R^2} + 2c_p^{-1}\left[M^{-1/2}|\zeta^\top W \mu| + |1 - 2|\mathcal{N}|/M| \cdot |\mu^\top W \zeta_{M+1}| + |M^{-1/2}\zeta^\top W \zeta_{M+1}|\right]\Bigg)$$

$$\le p + 2\exp(-c_p^2 M) + \mathbb{P}(\mu^\top W \mu - \mathbb{E}[\mu^\top W \mu] < -\frac{1}{2}\rho \cdot \frac{\tilde{R}^2}{R^2}) + \mathbb{P}\left(2c_p^{-1}M^{-1/2}|\zeta^\top W \mu| > \frac{\rho \tilde{R}^2}{8R^2}\right)$$

$$+ \mathbb{P}\left(|1 - 2|\mathcal{N}|/M| \cdot |\mu^\top W \zeta_{M+1}| \ge \frac{\rho \tilde{R}^2}{8R^2}\right) + \mathbb{P}\left(|M^{-1/2}\zeta^\top W \zeta_{M+1}| \ge \frac{\rho \tilde{R}^2}{8R^2}\right). \tag{41}$$

Inequality $(i)$ uses (34). From here the proof is exactly the same as in the clean case. For the first term, we use similar arguments used derive (36). To apply Lemma C.2, we need to examine the quantity $\frac{td}{\tilde{R}^2\|W\|_F}$ when $t = \rho\tilde{R}^2/2R^2$: we have,

$$\frac{td}{\tilde{R}^2\|W\|_F} = \frac{\rho d}{2R^2\|W\|_F} \ge \frac{\rho d}{2R^2 \cdot 2\sqrt{d}R^{-2}} = \frac{\rho\sqrt{d}}{4}. \tag{42}$$

Again if $\rho\sqrt{d}/4 > 1$ then we have by Lemma C.2,

$$\mathbb{P}\left(\mu^\top W \mu - \mathbb{E}[\mu^\top W \mu] < -\frac{\rho\tilde{R}^2}{2R^2}\right) \le 2\exp\left(-c\min\left(\frac{d^2}{\tilde{R}^4\|W\|_F^2} \cdot \frac{\rho^2\tilde{R}^4}{4R^4}, \frac{d}{\tilde{R}^2\|W\|_F} \cdot \frac{\rho\tilde{R}^2}{2R^2}\right)\right)$$

$$\le 2\exp\left(-\frac{cd\rho}{2\|W\|_F R^2}\right)$$

$$\overset{(i)}{\le} 2\exp\left(-\frac{c\rho\sqrt{d}}{4}\right). \tag{43}$$

Inequality $(i)$ uses (42). Note that if $\rho\sqrt{d}/4 \le 1$ then the above bound still holds since $\mathbb{P}(\cdot) \le 1$ is a trivial inequality.

As for the $\mu^\top W \zeta_{M+1}$ and $\zeta^\top W \mu$ terms, similarly as to the analysis used to derive (37) via Lemma C.4, we have,

$$\mathbb{P}\left(2c_p^{-1}|\mu^\top W \zeta_{M+1}| \ge \frac{\rho\tilde{R}^2}{8R^2}\right) = \mathbb{P}\left(|\mu^\top W \Lambda^{1/2} g| > \frac{c_p\rho\tilde{R}^2}{16R^2}\right) \qquad (g \sim \mathsf{N}(0, I))$$

$$\le 2\exp\left(-\frac{cc_p\rho\tilde{R}}{32\|\Lambda^{1/2}\|}\right) \tag{44}$$

Note that $|1 - 2|\mathcal{N}|/M| \leq 1$ always so this takes care of the term involving $\mu^\top W \zeta_{M+1}$. Since the analysis used to derive (37) only relies upon upper bounds for $\|W\|_F = \|W^\top\|_F$, and since $M \geq 1$ and $c_p \in (0, 1/2]$, the same analysis holds for the term $|\zeta^\top W \mu|$.

For the $\zeta^\top W \zeta_{M+1}$ term, the same analysis used for (38) applies here as well, since

$$\mathbb{P}\left(|M^{-1/2}\zeta^\top W \zeta_{M+1}| \geq \frac{\rho \tilde{R}^2}{8R^2}\right) = \mathbb{P}\left(|\zeta^\top W \zeta_{M+1}| \geq \frac{\rho M^{1/2}\tilde{R}^2}{8R^2}\right)$$

$$\leq 2\exp\left(-\frac{c\rho M^{1/2}\tilde{R}^2}{16\|\Lambda\|_2\sqrt{d}}\right) \tag{45}$$

Plugging (43), (44), and (45) into (41) we get

$$R(W_{\mathsf{MM}}) \leq p + 2\exp(-c_p^2 M) + 2\exp\left(-\frac{c\rho\sqrt{d}}{4}\right) + 4\exp\left(-\frac{cc_p\rho\tilde{R}}{32\|\Lambda^{1/2}\|}\right) + 2\exp\left(-\frac{c\rho M^{1/2}\tilde{R}^2}{16\|\Lambda\|_2\sqrt{d}}\right).$$

By adjusting the constant $c$ to depend on $c_p$, this completes the proof of Theorem 4.1.

## D    PROOF OF THEOREM 4.2

Recall that we assume that $\Lambda = I$ in this section, and for simplicity denote $W = W_{\mathsf{MM}}$. Our goal is to show that for each $k$,

$$\hat{\mu}^\top W y_k x_k > 0.$$

We have:

$$\hat{\mu}^\top W y_k x_k = \left(\frac{1}{M}\sum_{i=1}^{M} y_i x_i\right)^\top W y_k x_k$$

$$= \frac{1}{M}\left[x_k^\top W x_k + \sum_{i \neq k} y_i y_k x_i^\top W x_k\right].$$

Our goal here will be to show that the first quantity $x_k^\top W x_k$ is large and positive and dominates the second term involving $x_i^\top W x_k$. By definition we have

$$x_k^\top W x_k = (\mu + y_k z_k)^\top W(\mu + y_k z_k)$$
$$= \mu^\top W \mu + y_k z_k^\top W \mu + \mu^\top W y_k z_k + z_k^\top W z_k. \tag{46}$$

In particular we have the identities

$$\{\hat{\mu}^\top W y_k x_k < 0\}$$
$$= \{x_k^\top W x_k < -\sum_{i \neq k} y_i y_k x_i^\top W x_k\}$$
$$= \{z_k^\top W z_k - \mathrm{tr}(W) < -\mathrm{tr}(W) - y_k z_k^\top W \mu - \mu^\top W y_k z_k - \mu^\top W \mu - \sum_{i \neq k} y_i y_k x_i^\top W x_k\}$$
$$\subset \{z_k^\top W z_k - \mathrm{tr}(W) < -\frac{1}{2}\mathrm{tr}(W)\} \cup \{|z_k^\top W \mu| > \frac{1}{8}\mathrm{tr}(W)\}$$
$$\cup \{|\mu^\top W z_k| > \frac{1}{8}\mathrm{tr}(W)\} \cup \{\mu^\top W \mu < -\frac{1}{8}\mathrm{tr}(W)\} \cup \left\{\left|\sum_{i \neq k} y_i y_k x_i^\top W x_k\right| > \frac{1}{8}\mathrm{tr}(W)\right\}. \tag{47}$$

We will therefore proceed by bounding the probability each of these events. We will use the same notation from the proof of Theorem 4.1 presented in Appendix C, whereby $i \in \mathcal{C}$ refers to clean examples $(x_i, y_i)$ with $\tilde{y}_i = y_i$, while $i \in \mathcal{N}$ means $y_i = -\tilde{y}_i$.

## D.1 $\sum_{i \neq k} x_i^\top W x_k$ TERM.

We have,

$$\sum_{i \neq k} y_i x_i = \sum_{i \in \mathcal{C},\, i \neq k} y_i x_i + \sum_{i \in \mathcal{N},\, i \neq k} y_i x_i$$

$$= \sum_{i \in \mathcal{C},\, i \neq k} (\mu + \tilde{y}_k z_k) + \sum_{i \in \mathcal{N},\, i \neq k} (-\mu - \tilde{y}_k z_k)$$

$$= (|\mathcal{C} \setminus \{k\}| - |\mathcal{N} \setminus \{k\}|)\mu + \sum_{i \in \mathcal{C},\, i \neq k} \tilde{y}_k z_k - \sum_{i \in \mathcal{N},\, i \neq k} \tilde{y}_k z_k.$$

Now since the label-flipping noise is independent of $z_k$, the random variables $\tilde{y}_k z_k$ are i.i.d. standard normals. Therefore there is $g_k \sim \mathsf{N}(0, I_d)$ such that

$$\sum_{i \in \mathcal{C},\, i \neq k} \tilde{y}_k z_k - \sum_{i \in \mathcal{N},\, i \neq k} \tilde{y}_k z_k = \sqrt{M-1} g_k.$$

If we denote $N_k := |\mathcal{C} \setminus \{k\}| - |\mathcal{N} \setminus \{k\}|$ then clearly $|N_k| \leq M$ and thus we have

$$\sum_{i \neq k} y_i x_i = N_k \mu + \sqrt{M-1} g_k,$$

where $|N_k| \leq M$ and $g_k$ is a standard normal and $g_k$ is independent of $z_k$ and $y_k$. Therefore

$$\left| \sum_{i \neq k} y_i x_i^\top W y_k x_k \right| = \left| (N_k \mu + \sqrt{M-1} g_k)^\top W (\tilde{y}_k y_k \mu + y_k z_k) \right|$$

$$\leq M|\mu^\top W \mu| + \sqrt{M}|g_k^\top W \mu| + M|\mu^\top W z_k| + \sqrt{M}|g_k^\top W z_k|. \qquad (48)$$

In particular we have

$$\mathbb{P}\left( \left| \sum_{i \neq k} y_i x_i^\top W y_k x_k \right| > \frac{\operatorname{tr}(W)}{8} \right) \leq \mathbb{P}\left( M|\mu^\top W \mu| > \frac{\operatorname{tr}(W)}{32} \right)$$

$$+ \mathbb{P}\left( \sqrt{M}|g_k^\top W \mu| > \frac{\operatorname{tr}(W)}{32} \right) + \mathbb{P}\left( M|\mu^\top W z_k| > \frac{\operatorname{tr}(W)}{32} \right) + \mathbb{P}\left( \sqrt{M}|g_k^\top W z_k| > \frac{\operatorname{tr}(W)}{32} \right). \qquad (49)$$

We'll proceed by bounding each of these terms in sequence.

$\mu^\top W \mu$ **term.** Let $L$ be an integer (we will take $L = M$ in this proof but in a later proof we will take $L = 1$). Since $\operatorname{tr}(W) > 0$ by Lemma B.1, we have

$$\left\{ |L\mu^\top W \mu| > \frac{\operatorname{tr}(W)}{32} \right\} = \left\{ \left| \mu^\top W \mu - \frac{\tilde{R}^2 \operatorname{tr}(W)}{d} \right| + \frac{\tilde{R}^2 \operatorname{tr}(W)}{d} > \frac{\operatorname{tr}(W)}{32L} \right\}$$

This means

$$\mathbb{P}\left( |L\mu^\top W \mu| > \frac{\operatorname{tr}(W)}{32} \right) \leq \mathbb{P}\left( \left| \mu^\top W \mu - \frac{\tilde{R}^2 \operatorname{tr}(W)}{d} \right| > \frac{\operatorname{tr}(W)}{32L} \left( 1 - \frac{32\tilde{R}^2 L}{d} \right) \right).$$

Assume for now $d > 128\rho^{-1} \tilde{R}^2 L$ so that $32\tilde{R}^2 L/d < 1/2$. We then have by Lemma C.2,

$$\mathbb{P}\left( |L\mu^\top W \mu| > \frac{\operatorname{tr}(W)}{32} \right) \leq \mathbb{P}\left( \left| \mu^\top W \mu - \frac{\tilde{R}^2 \operatorname{tr}(W)}{d} \right| > \frac{\operatorname{tr}(W)}{64L} \right)$$

$$\leq 2\exp\left( -c \min\left( \frac{d^2}{\tilde{R}^4 \|W\|_F^2} \left( \frac{\operatorname{tr}(W)}{64L} \right)^2, \frac{d}{\tilde{R}^2 \|W\|_F} \left( \frac{\operatorname{tr}(W)}{64L} \right) \right) \right).$$

As for the quantity appearing in the minimum we have by Lemma B.1

$$\frac{d \operatorname{tr}(W)}{64L} \cdot \frac{1}{\tilde{R}^2} \geq \frac{d}{64L} \cdot \frac{\rho d}{R^2} \cdot \frac{R^2}{2\tilde{R}^2 \sqrt{d}} = \frac{\rho d^{3/2}}{128L\tilde{R}^2} \geq \frac{\rho d}{128L\tilde{R}^2}.$$

We therefore have

$$\mathbb{P}\left(|L\mu^\top W\mu| > \frac{\mathrm{tr}(W)}{32}\right) \le 2\exp\left(-\frac{c\rho d}{128L\tilde{R}^2}\right). \tag{50}$$

We see that the case $d \le 128\rho^{-1}\tilde{R}^2 L$ results in the same inequality to hold (albeit vacuously). This completes the proof for the $\mu^\top W\mu$ term in (49) by taking $L = M$.

$g_k^\top W\mu$ **and** $\mu^\top W z_k$ **terms.** The terms $\mu^\top W g_k$ and $z_k^\top W\mu$ can be controlled using Lemma C.4. For an integer $L \in \mathbb{N}$ we have

$$\mathbb{P}\left(L|\mu^\top W g_k| \ge \frac{\mathrm{tr}(W)}{32}\right) \le 2\exp\left(-\frac{c\sqrt{d}}{\tilde{R}\|W\|_F} \cdot \frac{\mathrm{tr}(W)}{8L}\right).$$

We have

$$\frac{\sqrt{d}}{\tilde{R}\|W\|_F} \cdot \frac{\mathrm{tr}(W)}{32L} \ge \frac{R^2\sqrt{d}}{\tilde{R}\cdot 2\sqrt{d}} \cdot \frac{\rho d}{32R^2 L} = \frac{\rho d}{64\tilde{R}L}.$$

Note that the above argument applies to $W^\top$ as well as $W$, so that we have

$$\mathbb{P}\left(L|g_k^\top W\mu| \ge \frac{\mathrm{tr}(W)}{32}\right) \vee \mathbb{P}\left(L|\mu^\top W g_k| \ge \frac{\mathrm{tr}(W)}{32}\right) \le 2\exp\left(-\frac{c\rho d}{64\tilde{R}L}\right). \tag{51}$$

In particular, we have

$$\mathbb{P}\left(\sqrt{M}|g_k^\top W\mu| \ge \frac{\mathrm{tr}(W)}{32}\right) \le 2\exp\left(-\frac{c\rho d}{64\tilde{R}\sqrt{M}}\right),$$
$$\mathbb{P}\left(M|\mu^\top W z_k| \ge \frac{\mathrm{tr}(W)}{32}\right) \le 2\exp\left(-\frac{c\rho d}{64\tilde{R}M}\right). \tag{52}$$

$g_k^\top W z_k$ **term.** The last term $g_k^\top W z_k$ is a Gaussian chaos random variable. By Vershynin (2018, Lemma 6.2.2) this random variable is sub-exponential with sub-exponential norm at most $c\|W\|_F$. Therefore we have

$$\mathbb{P}(\sqrt{M}|g_k^\top W z_k| > \frac{\mathrm{tr}(W)}{32}) = \mathbb{P}(|g_k^\top W z_k| > \frac{\mathrm{tr}(W)}{32\sqrt{M}}) \le 2\exp\left(-c\frac{\mathrm{tr}(W)}{32\sqrt{M}\|W\|_F}\right).$$

Since $\mathrm{tr}(W)/\|W\|_F \ge \rho\sqrt{d}/2$ by Lemma B.1 this implies

$$\mathbb{P}(\sqrt{M}|g_k^\top W z_k| > \frac{\mathrm{tr}(W)}{32}) \le 2\exp\left(-\frac{c\rho\sqrt{d}}{64\sqrt{M}}\right). \tag{53}$$

**Putting it all together.** Continuing from (49) we can use (50), (52) and (53) to get

$$\mathbb{P}\left(\left|\sum_{i\ne k} y_i x_i^\top W y_k x_k\right| > \frac{\mathrm{tr}(W)}{8}\right) \le 2\exp\left(-\frac{c\rho d}{128M\tilde{R}^2}\right)$$
$$+ 2\exp\left(-\frac{c\rho d}{64\tilde{R}\sqrt{M}}\right) + 2\exp\left(-\frac{c\rho d}{64M\tilde{R}}\right) + 2\exp\left(-\frac{c\rho\sqrt{d}}{64\sqrt{M}}\right). \tag{54}$$

### D.2  $x_k^\top W x_k$ TERM.

It remains to bound the probability of the first three events in (47).

$z_k^\top W z_k$ **term.** First note that since $z_k$ is a standard Gaussian, by a standard Hanson-Wright inequality (Vershynin, 2018, Lemma 6.2.1) we have for some constant $0 < c < 1$, for any $t \geq 0$,

$$\mathbb{P}(|z_k^\top W z_k - \mathrm{tr}(W)| \geq t) \leq 2 \exp\left(-c \min\left(\frac{t^2}{\|W\|_F^2}, \frac{t}{\|W\|_F}\right)\right).$$

So for $t = \frac{1}{2}\mathrm{tr}(W)$ we have

$$\frac{t}{\|W\|_F} = \frac{\mathrm{tr}(W)}{2\|W\|_F} \geq \frac{\rho d}{R^2} \cdot \frac{R^2}{4\sqrt{d}} = \frac{\rho\sqrt{d}}{4}.$$

This means that if $\rho\sqrt{d}/4 \geq 1$ we have by Lemma B.1

$$\mathbb{P}\left(|z_k^\top W z_k - \mathrm{tr}(W)| \geq \frac{1}{2}\mathrm{tr}(W)\right) \leq 2 \exp\left(-c\rho\sqrt{d}/4\right). \tag{55}$$

Since the same inequality trivially holds if $\rho\sqrt{d}/4 < 1$ (by potentially taking $c$ to be a smaller constant), this completes this term.

$\mu^\top W \mu$ **term.** This is covered by (50) with $L = 1$:

$$\mathbb{P}\left(|\mu^\top W \mu| > \frac{\mathrm{tr}(W)}{32}\right) \leq 2 \exp\left(-\frac{c\rho d}{128\tilde{R}^2}\right). \tag{56}$$

$z_k^\top W \mu$ **terms.** Since $g_k$ and $z_k$ are i.i.d., this is covered by (51) with $L = 1$:

$$\mathbb{P}\left(|z_k^\top W \mu| \geq \frac{\mathrm{tr}(W)}{32}\right) \vee \mathbb{P}\left(|\mu^\top W z_k| \geq \frac{\mathrm{tr}(W)}{32}\right) \leq 2 \exp\left(-\frac{c\rho d}{64\tilde{R}}\right). \tag{57}$$

**Putting it all together.** Continuing from (47) we have

$$\mathbb{P}\left(\widehat{\mu}^\top W y_k x_k < 0\right)$$
$$\leq \mathbb{P}\left(z_k^\top W z_k - \mathrm{tr}(W) < -\frac{1}{2}\mathrm{tr}(W)\right) + \mathbb{P}\left(|z_k^\top W \mu| > \frac{1}{8}\mathrm{tr}(W)\right)$$
$$+ \mathbb{P}\left(|\mu^\top W z_k| > \frac{1}{8}\mathrm{tr}(W)\right) + \mathbb{P}\left(\mu^\top W \mu < -\frac{1}{8}\mathrm{tr}(W)\right) + \mathbb{P}\left(\left|\sum_{i \neq k} y_i y_k x_i^\top W x_k\right| > \frac{1}{8}\mathrm{tr}(W)\right)$$
$$\leq 2 \exp\left(-\frac{c\rho\sqrt{d}}{4}\right) + 4 \exp\left(-\frac{c\rho d}{64\tilde{R}}\right) + 2 \exp\left(-\frac{c\rho d}{128M\tilde{R}^2}\right)$$
$$+ 2 \exp\left(-\frac{c\rho d}{64\tilde{R}\sqrt{M}}\right) + 2 \exp\left(-\frac{c\rho d}{64M\tilde{R}}\right) + 2 \exp\left(-\frac{c\rho\sqrt{d}}{64\sqrt{M}}\right)$$
$$\leq 4 \exp\left(-\frac{c\rho\sqrt{d}}{64\sqrt{M}}\right) + 8 \exp\left(-\frac{c\rho d}{128M(\tilde{R}^2 \vee \tilde{R})}\right).$$

Using a union bound this allows for us to conclude

$$\mathbb{P}\left(\exists k \in [M] \text{ s.t. } \widehat{\mu}^\top W y_k x_k < 0\right) \leq 4M \exp\left(-\frac{c\rho\sqrt{d}}{64\sqrt{M}}\right) + 8M \exp\left(-\frac{c\rho d}{128M(\tilde{R}^2 \vee \tilde{R})}\right),$$

as claimed in the theorem.

## E  EXPERIMENT DETAILS

We describe here the experimental setup for Figures 2 and 3; code is available on Github.[3]

---

[3]https://github.com/spencerfrei/icl_classification

We pre-train models using standard full-batch gradient descent on the logistic loss with $R = 5\sqrt{d}$, $N = 40$, learning rate $\eta = 0.01$, for 300 steps from a zero initialization, using PyTorch. The in-context training accuracy is measured using the definition of training accuracy from Theorem 4.2: namely, we look at what proportion of the in-context examples (training data) that is accurately predicted with the model $\widehat{y}(E_\tau^{1:M}; W)$, where $W$ is the trained transformer, for a single task $\tau$, i.e. averaging $\mathbb{1}(y_k = \text{sign}(y(E_\tau^{1:M}(x_k); W)))$ over $k = 1, ..., M$. The in-context test accuracy is computed by measuring whether $\text{sign}(\widehat{y}(E_\tau^{1:M}(x_{M+1}); W)) = y_{M+1}$. We then average over 2500 tasks, and we plot this average with error bars corresponding to one standard error over these 2500 numbers. All computations can be run within an hour on high-quality CPU, although we used an NVIDIA RTX 3500 Ada which helped speed up the computations for the $B = d \geq 1000$ setting.

