# OpenReview forum: "Trained Transformer Classifiers Generalize and Exhibit Benign Overfitting In-Context"
_ICLR.cc/2025/Conference — ICLR 2025 Poster_

### Official Review · Reviewer_F6y8 · 2024-11-01

**Soundness:** 3
**Presentation:** 3
**Contribution:** 3
**Rating:** 6
**Confidence:** 3

**Summary:**

This paper studies in-context learning of linear classification tasks on linear transformers. The authors characterize how many pre-training tasks and in-context samples are needed for generalization, with an analysis of the implicit regularization of gradient descent. They show that after pre-training the transformer can tolerate smaller SNR than those tasks on which it was trained. Additionally, they show these trained transformers can memorize all in-context samples yet still generalize near-optimally when the in-context samples’ labels are flipped.

**Strengths:**

1. The study of in-context learning is important, and the investigation on linear classification setting is novel.
2. The results of benign overfitting of transformer, to the reviewer’s knowledge, is new.
3. The paper is well-written and easy to follow.

**Weaknesses:**

1. It would have been interesting to discuss the connection of task complexity results under linear classification to Wu et al. [1]. In addition, briefly mentioning the importance of investigating linear classification instead of regression in the paper would be beneficial.
2. Can you explain why it would be interesting to consider a different signal-to-noise ratio R\tilde at test time? Analyzing the difference between M and N seems to be more interesting.
3. The reviewer personally would like to see consistent empirical observation with simple experiments (such as a simplified version of Raventós et al [2]) but understand this might be out of the scope of this work. Maybe the author can consider mentioning it as a limitation.

[1] How Many Pretraining Tasks Are Needed for In-Context Learning of Linear Regression? Wu et al. ICLR 2024
[2] Pretraining task diversity and the emergence of non-Bayesian in-context learning for regression. Raventós et al. NeurIPS 2023

**Questions:**

1. At row 72, 73 → ‘even when pre-training on simple and easy-to-learn datasets, the transformer can generalize on more complex tasks.’ What does easy and complex task mean in this context? Does it mean the signal-to-noise ratio?

Minor:
1. At line 331, a typo of ‘sufficiently’.

---

> ### Author Response · Authors · 2024-11-21
>
> Thank you for the constructive feedback. We address specific points:
>
> **Task Complexity Compared to Wu et al.**: This is a great question.  Our setting is not directly comparable since in their setting the covariance matrix of the data ($E\[xx^T\]$) does not depend on the underlying predictor, and so their bound in Theorem 4.1 shows a task complexity which depends explicitly on the trace of the data covariance matrix, and as this trace is larger the problem is harder (more variance, harder to learn).  In the classification setting with class-conditional Gaussian data that we consider, the data covariance matrix takes the form $E\[xx^\\top\] \= E\[(y\\mu \+ z)(y\\mu \+ z^\\top)\] \= (R^2/d) I \+ \\Lambda$, so its trace gets larger when either R gets larger (which makes learning easier) *or* when the cluster noise covariance $\\Lambda$ gets larger (which makes learning harder).  It’s thus hard to directly compare these two settings.
>
> **Motivation of classification:** Our goal is to eventually understand what’s happening in more real-world transformer models, which rely upon training on the cross-entropy loss and when the data comes from discrete tokens.  We think a useful step towards this goal is to begin to understand the behavior of transformers which are trained on classification losses in settings which we understand well, like class-conditional Gaussians.
>
> **Signal-to-Noise Ratio**: Different $\tilde R$ at test time is interesting because it shows transformers can generalize to more challenging settings than seen during training.  This is particularly relevant for real-world applications where test conditions may be more difficult than training.  We also think it’s notable that the transformer has *very* different behavior when $\\tilde R$ is small, medium, or large (see our new Figures 2 and 3).
>
> We agree that understanding M vs. N is also an important distinction, and although this work provides a fairly complete picture of the dependence on M when N is large, we think it’s an interesting direction for future research to understand what happens when N is small.
>
> **Question re: easy/complex:** Sorry for the lack of clarity here.  By “easy” we were referring to both high SNR as well as lack of label-flipping noise, and by “complex” we meant low SNR and the presence of label-flipping noise.
>
> **Experiments**: We have added new experiments to the paper, see Figures 2 and 3, which reveal the importance of $\\tilde R$ and high-dimensionality for benign overfitting, as well as the role of the number of pre-training tasks $B$.  We are happy to provide our code or run further experiments if the reviewer is interested!

---

> > ### Comment · Reviewer_F6y8 · 2024-11-22
> >
> > Thank you for the clarifications.
> > Regarding the 'motivation of classification', I agree that classification is closer to real-world examples and that's what one would naturally do, so I was initially curious about why all prior works start from (and only consider) regression, is there a specific reason/challenge that prevents them from analyzing a classification loss? This is something not clearly mentioned in the current version IIRC.
> > Regarding the task complexity part, I propose to clarify it in the future version of the paper.
> > Overall, I still lean on accept and will maintain my initial score.

---

> > > ### Author Response · Authors · 2024-11-25
> > >
> > > That is a good question.  We too are unsure of why there are many prior works which looked at regression but, to our knowledge, no prior works investigated the linear classification setting.  One potential explanation is that for classification losses, there are many minimizers of the loss and different minimizers may have different downstream properties.  Understanding which minimizer is chosen (namely, the implicit regularization of gradient descent) is thus needed to understand these downstream properties (like in-context learning ability), which forms the core of our proof.  We've revised the PDF to add some discussion on this around lines 142-149.

---

### Official Review · Reviewer_kNQi · 2024-11-02

**Soundness:** 3
**Presentation:** 3
**Contribution:** 2
**Rating:** 6
**Confidence:** 3

**Summary:**

The paper investigated the behavior of linear transformer models trained on random classification tasks, extending the existing work on linear regression tasks. It studied how linear transformers can generalize and exhibit _benign overfitting_, meaning that even if a model memorizes noisy training examples, it still performs well on test examples. This paper is completely theoretical, and no empirical evidence was given.

**Strengths:**

- This work theoretically analyzed how many pre-training tasks and in-context examples are needed for linear transformers to effectively generalize in linear classification tasks.
- This work studied the phenomenon of benign overfitting in this context and bounded the generalization error.
- The paper is solid and well presented. All the symbols are properly defined. The assumptions and theoretical results are clearly stated. (I did not carefully check the proofs.)
- The paper is well contextualized. The author mentioned related work (that I'm not familiar with) and discussed future work directions, which helped me position this work.

**Weaknesses:**

My biggest concern is the same for all theoretical work on simple models (e.g., single-layer linear transformer) with strong data assumptions (e.g., class-conditional Gaussian mixture). It is perfectly fine to start with a simple model with strong assumptions, as long as it is a good proxy for real-world problems or a stepping stone to understanding more complex problems. However, we need either theoretical or empirical evidence showing it is the case. In the current paper, there is a lack of discussion (or references) on whether or to what extent real-world data satisfies the data assumptions. It is unclear whether the theoretical results can be generalized to more complex models. Overall, I'd like to know how the theoretical results can guide our practice, which is not completely clear in the current version.

**Questions:**

In Figure 1, the author used a natural language example to show the phenomenon of benign overfitting. However, I don't think it satisfies the data assumption. Is there an example/illustration that precisely satisfies the data/model assumptions?

---

> ### Author Response · Authors · 2024-11-21
>
> We appreciate the positive feedback on our work. Regarding the key concern on the data assumptions: we agree that the assumptions we make are relatively strong.  However, we think these assumptions are reasonable for a theoretical work that gives a precise analysis of the effects of all of the core ingredients to the transformer training pipeline: the implicit regularization effect of gradient descent in pre-training; the number of pre-training tasks; the number of examples per pre-training task; the signal-to-noise ratio of the pre-training tasks; the number of in-context examples as test time; the signal-to-noise ratio of the test-time tasks; and the effect of label noise at test time.  To our knowledge, no prior work has captured all of these aspects of pre-training/testing transformers simultaneously.  Moreover, we discovered the possibility of benign overfitting in the forward pass of the transformer, which has not appeared in prior work.
>
> We think this work paves the way for future works which we hope can develop as fine-grained of a characterization of the interplay between data, architecture, and optimizers as we did but for transformers with more complex architectures and realistic data distributions.  As we mention in the discussion, we believe it should be possible to extend our analysis to non-convex linear transformer architectures which are homogeneous in the parameters, since the KKT condition analysis approach can work in that setting as well.
>
> Regarding the question: we’ve revised our manuscript with experiments which demonstrate a setting which satisfies all of our assumptions and exhibits the type of behaviors that our theory suggests—see Figures 2 and 3 with details on the experiments in lines 377-400 and Appendix E.  Namely, benign overfitting in-context occurs when the data is sufficiently high-dimensional and $\tilde R$ is in a sweet spot, and test error improves with greater pre-training tasks, especially when $\tilde R$ isn’t too large.

---

### Official Review · Reviewer_5w9f · 2024-11-04

**Soundness:** 3
**Presentation:** 2
**Contribution:** 3
**Rating:** 6
**Confidence:** 3

**Summary:**

This work presents a sample complexity analysis for in-context learning of class-conditional Gaussian mixtures with a restricted linear attention transformer model (single layer). It uses the results established on how SGD on convex-linear transformer models has a bias toward maximum-margin solution (in direction). The work thus assumes the KKT conditions are satisfied at the SGD solution and uses them to quantify the sample requirement to achieve a small test error. It shows how --by choice of specific model and data parameters-- one can achieve benign overfitting for in-context training, a phenomenon in which noisy training examples are exactly memorized, yet the test accuracy remains near-optimal.

**Strengths:**

- The work references and uses prior work to great effect. It builds up on ideas established around the implicit regularization effect of the optimization algorithms used to train transformer models. SGD-based algorithms applied to convex-linear transformers have a directional bias towards maximum-margin solutions. Hence once can apply techniques used to study such solutions to derive generalization bounds such as the one presented in 4.1 and 4.2.

**Weaknesses:**

- Readability: The paper is very dense and in parts hard to follow, requiring multiple revisits to earlier sections for (non-standard) notation and technical definitions. This, unfortunately, severely cuts into the readability of the paper. Adding a table of notation either in the paper or the appendix can be helpful.

- Analysis: There is no empirical evaluation of the tightness of derived bounds. IMO the derived bounds are to be understood in the limit and to prove the theoretical possibility of benign over-fitting. They might be too lose in practice. Adding an study in which $M$ or $d$ can be varied can help establish the empirical tightness of the bound.

- Scope: The work only focuses on a specific restricted linear model. It defers analysis on non-convex linear or softmax-based attention models to future work.

- Assumptions: Even with the SGD-based solutions being directionally biased towards max-margin solutions, it doesn’t necessarily follow that the KKT conditions are satisfied in real implementations with early stopping or partial convergence. It’s also a leap to assume this (line of) theoretical analysis can be extended to other settings beyond the restricted or convex linear case addressed in the paper. IMO these assumption should be studied/validated under realistic training settings. Adding such analysis to the paper can clear some of these points.

**Questions:**

- Could tighter bounds be established for stochastic (Gibbs) classifiers? This would be stochasticity in the loss that is not a result of label-flips on the test set, but is a result of stochastic classifier decision.


- Line 377 breaks a formula.

---

> ### Author Response · Authors · 2024-11-21
>
> Thank you for the detailed review. We address your concerns:
>
> **Readability**: We appreciate the feedback about density and notation. We have added a comprehensive notation table in the appendix, see Table 1 and the added reference in line 138\.
>
> **Experiments for early stopping**: The reviewer raises an important point about KKT conditions not being satisfied under early stopping. Our analysis focuses on the theoretical limits of the limiting behavior of the transformer, similar to other works analyzing implicit regularization (citation).  We agree that studying partial convergence would be valuable , and we have added experiments which investigate empirically what happens in early-stopped networks, which broadly shows that early-stopped networks have similar behavior to what our theory suggests (see Figures 2 and 3, lines 377-400 in the revised paper, and Appendix E for details on experiments).
>
> **Regarding stochasticity and our focus on KKT conditions**: To clarify, our analysis holds for the limiting behavior of both GD and SGD with a constant step size, due to [Nacson et al. 2019](https://proceedings.mlr.press/v89/nacson19a.html).  In general we think there is a lot of value in investigating if generalization in neural networks can be explained solely via an analysis of the implicit regularization of the optimizer—if this is possible, then we have persuasive evidence for *why* this generalization behavior occurs.  If it isn’t possible, then it points to other aspects of the neural net training process which we should focus on instead.

---

> > ### Comment · Reviewer_5w9f · 2024-11-25
> > **Review on Updates**
> >
> > The authors have added an experiment to the paper the studies the effects of early stopping (finite examples) in the observed overfitting effects. I still think a larger detailed study of KKT conditions on realistic convergence settings is likely useful for this line of analysis, especially since the authors argue for using the regularization effects of the optimization algorithm in future studies.
> >
> > I have updated my assessment to reflect the added experiments and improvements to the paper.

---

### Official Review · Reviewer_dEuQ · 2024-11-04

**Soundness:** 3
**Presentation:** 3
**Contribution:** 3
**Rating:** 6
**Confidence:** 3

**Summary:**

This paper studies the ability of linear transformers to perform in-context learning for linear classification tasks. The authors consider a simplified, convex linear transformer architecture trained on a series of random, class-conditional Gaussian mixture models.

The paper investigates two key aspects:

1. the number of pre-training tasks required for the transformer to generalize well at test time, even when the test data has lower signal-to-noise ratios (SNR) and label-flipping noise not present in the pre-training data.

2. the phenomenon of “benign overfitting in-context,” where the transformer memorizes noisy in-context training examples while still achieving near-optimal generalization for clean test examples.

The authors claim this work to be the first theoretical analysis of in-context learning for linear classification using linear transformers, and also the first demonstration of benign overfitting capabilities.

This paper makes valuable theoretical contributions to the understanding of in-context learning in transformers, highlighting their generalization capabilities and the intriguing phenomenon of benign overfitting. While the simplified model and assumptions might limit direct practical implications, the paper opens up interesting avenues for future research on the theoretical foundations of in-context learning.

**Strengths:**

Rigorous theoretical analysis: The paper offers a detailed analysis of the implicit regularization of gradient descent during pre-training and leverages the max-margin framework to derive generalization guarantees.

Novel insights: The paper reveals the intriguing ability of the trained transformer to generalize under lower SNR and label-flipping noise at test time, even when such conditions are absent during pre-training.

Demonstration of benign overfitting: The paper provides theoretical and experimental evidence for the previously unobserved phenomenon of benign overfitting in transformers.

This paper presents novel theoretical insights into the in-context learning capabilities of transformers for linear classification tasks. The demonstration of benign overfitting is particularly noteworthy. While the simplified assumptions and focus on a convex architecture limit the direct applicability of the findings, the paper offers a valuable foundation for future research in understanding in-context learning in more realistic transformer settings.

**Weaknesses:**

Simplified architecture: The analysis relies on a 1-layer linear transformer model, which is a simplification of standard multi-layer transformer models with softmax-based attention. It is unclear whether about the applicability of the results to more complex transformer architectures.

Noise-free pre-training: The requirement of clean data during pre-training restricts the practicality of the approach in noisy environments.

Strong assumptions on data distribution: The paper focuses on class-conditional Gaussian mixture models, which might not be representative of the complex data distributions encountered in natural language processing tasks.

Absence of Numerical Validation: While the theoretical analysis is valuable, the paper lacks numerical experiments to validate the theoretical findings. Providing empirical evidence to support the theoretical claims, and demonstrating how the results generalize to more realistic settings, would enhance the paper's impact and persuasiveness.

**Questions:**

Practical Implications: Recognizing the limitations of the simplified setting, can the authors discuss any potential practical insights for realistic problems?

Exploration of Alternative Architectures: How would the results generalize to more complex transformer models?

Pre-Training with Noisy Labels: The study assumes clean input data and labels during pre-training. How would the result change once noise is introduced in the pre-training data?

Numerical Validation of Theoretical Results: Have the authors tried conducting numerical experiments to validate your theoretical findings? It would be highly insightful to see how well the theoretical predictions align with empirical observations in both the simplified setting used for analysis and in more realistic settings involving complex transformer architectures and real-world datasets. For instance, could the authors present experimental results demonstrating the relationship between the number of pre-training tasks, the signal-to-noise ratio, and the generalization performance, as suggested by your theoretical bounds? Furthermore, showcasing empirical examples of benign overfitting in-context would be compelling evidence to support this key finding.

---

> ### Author Response · Authors · 2024-11-21
>
> We thank the reviewer for their thoughtful comments. We address the main points below:
>
> **Simplified Architecture**: While we analyze a 1-layer linear transformer, this simplification allows us to rigorously characterize fundamental properties of in-context learning that may extend to more complex architectures. Similar to how early theoretical works on neural networks focused on simplified architectures to establish foundational results (citation), our work provides a theoretical framework that can guide future analyses of more complex transformers. Moreover, prior works have utilized the same architecture to derive theoretical insights ([Wu et al. 2024](https://arxiv.org/abs/2310.08391), [Kim et al. 2024](https://arxiv.org/abs/2408.12186)).
>
> Regarding more complex architecture, as we mention in the conclusion we believe our techniques can be extended to more complex linear transformer architectures as the last-token prediction would still be homogeneous in the network parameters, enabling the usage of the implicit regularization of GD on the logistic loss for such architectures [(Lyu & Li, 2020\)](https://arxiv.org/abs/1906.05890).  We think an interesting but more challenging question is to extend our results to softmax attention architectures.
>
> **Numerical Validation**: We agree that empirical validation with GD would be valuable, and we have added these in the revised PDF; see Figures 2 and 3 with details on the experiments in lines 377-400 and Appendix E. The experiments broadly support our theoretical findings, demonstrating that benign overfitting in-context occurs for particular settings of $\\tilde R$, $d$, $M$, and $B$.
>
> However, like other foundational theoretical works in this area [(Von Oswald et al 2022](https://arxiv.org/abs/2212.07677), [Akyurek et al 2022](https://arxiv.org/abs/2211.15661), [Zhang et al 2023](https://arxiv.org/abs/2306.09927)), our focus is on establishing theoretical guarantees and understanding the principles of in-context learning. We believe our theoretical results are valuable on their own, as they:
>
> 1\. Provide precise analysis of the test-time performance of a pre-trained model and how this depends on the number of pre-training tasks, the signal-to-noise ratio, the number of in-context examples, and the noise rate.  To the best of our knowledge, prior works fail to capture all of these properties simultaneously, and none covered the linear classification setting.
>
> 2\. Demonstrate the possibility of benign overfitting in the forward pass of transformers: we believe this is a novel setting and finding which is of interest to theorists working on understanding LLM generalization.
>
> **Pre-training with noisy labels**: We think this is an interesting question.  We think the theoretical analysis of pre-training with noisy labels would be significantly different.  If we aimed to use the implicit regularization approach based on analyzing solutions to our equation (3), because in this setting there may not be a $U$ which satisfies the constraints of (3), and even if such a $U$ does exist then we would be investigating the behavior of a pre-trained transformer which has memorized noisy labels.  In this setting it is not clear that generalization is possible, and we believe a resolution to this question would be very challenging.  On the other hand, if we looked directly at the dynamics of GD, we conjecture that if GD is early-stopped before overfitting, we might expect similar behavior as to what we see for the max-margin in our paper.  We have added comments on this to our discussion in lines 506-513.

---

> > ### Comment · Reviewer_dEuQ · 2024-11-25
> >
> > Thank you for your response and addressing the concerns with the new updated results. I understand the challenge for establishing theoretical foundation for these problems. I'm happy with the update. I'm raising the score to 6.

---

### Meta-Review · Area_Chair_ZsZy · 2024-12-17

**Metareview:**

The paper provides a theoretical study of in-context learning in transformers. Specifically, the authors consider a single-layer linear attention transformer model, and assume that the pretraining data consists of linear classification problems sampled from a fixed task distribution. They then provide generalization bounds for in-context learning of the linear regression algorithm. The proof in particular relies on the implicit bias of gradient descent, as well as assumptions about the data. The authors derive generalization bounds in terms of the number of examples per task, number of pre-training tasks, signal-to-noise ratio. The authors also show that the models can generalize at test time to tasks with lower signal-to-noise ratio and also demonstrate benign overfitting, where the models memorize mislabeled examples but still generalize well. The authors also provide empirical support for their predictions.

Strengths:
- The paper derives the first generalization guarantees for in-context learning in transformers.
  + The bound depends on the number of pre-training tasks and in-context examples, which is qualitatively interesting.
- The authors show interesting generalization beyond pretraining task distribution: to higher SNR and benign overfitting.
+ The authors provide some empirical support for their derivations.

Weaknesses:
- The paper makes a number of strong assumptions
  + Very simple architecture: one linear attention layer. This architecture is very different from the transformer models used in practice.
  + Strong assumptions on the task / data distribution

Decision recommendation: The paper provides an interesting new analysis of in-context learning in a simplified theoretical setting. While the paper makes strong simplifying assumptions, I think it is still a valuable theoretical contribution and a starting point for understanding more practical models and data distributions. The reviewers are unanimous in accepting the paper. I recommend accepting the paper.

**Additional Comments On Reviewer Discussion:**

The reviewers are unanimous in accepting the paper. Generally, the reviewers were concerned by the strong theoretical assumptions made by the paper. The other concerns were the lack of empirical support and difficulties in readability. During the rebuttal, the authors engaged with the reviewers comments and provided clarifications as well as new experiments to support theoretical findings. As a result, multiple reviewers raised their scores.

---

### Decision · Program_Chairs · 2025-01-22

Accept (Poster)